# Cerebrospinal fluid reference proteins increase accuracy and interpretability of biomarkers for brain diseases

Linda Karlsson [1] ✉, Jacob Vogel[1,2], Ida Arvidsson [3], Kalle Åström[3], Shorena Janelidze[1], Kaj Blennow [4,5], Sebastian Palmqvist [1,6], Erik Stomrud[1,6], Niklas Mattsson-Carlgren[1,6] & Oskar Hansson [1,6] ✉

Cerebrospinal fluid (CSF) biomarkers reflect brain pathophysiology and are used extensively in translational research as well as in clinical practice for diagnosis of neurological diseases, e.g., Alzheimer's disease (AD). However, CSF biomarker concentrations may be influenced by non-disease related inter-individual variability. Here we use a data-driven approach to demonstrate the existence of inter-individual variability in mean standardized CSF protein levels. We show that these non-disease related differences cause many commonly reported CSF biomarkers to be highly correlated, thereby producing misleading results if not accounted for. To adjust for this inter-individual variability, we identified and evaluated high-performing reference proteins which improved the diagnostic accuracy of key CSF AD biomarkers. Our reference protein method attenuates the risk for false positive findings, and improves the sensitivity and specificity of CSF biomarkers, with broad implications for both research and clinical practice.

Neurodegenerative disorders and dementia are common and have increasing prevalence world-wide[1]. The need for precise and reliable diagnostic techniques to identify, examine and monitor these diseases is growing. One informative and cost-effective diagnostic technique is the measurement of protein concentrations in cerebrospinal fluid (CSF), here referred to as CSF biomarkers[2,3]. In Alzheimer's disease (AD), which is the most common neurodegenerative disease, CSF biomarkers are used in clinical practice as diagnostic tools[2]. Neuropathologically, AD is defined by the combined presence of amyloid(A)-β plaques and tau-neurofibrillary tangles. CSF biomarkers related to these pathologies include Aβ42 and soluble phosphorylated(P)-tau[4]. These CSF markers can substantially improve the diagnostic work-up of the disease, which is becoming increasingly important due to recent development of effective disease modifying-treatments for AD[5–7]. However, the use of CSF biomarkers may be complicated by inter-individual variability in certain non-disease related physiological phenomena, such as (but not limited to) subject-level variations in the transport rates of proteins from the brain parenchyma into the CSF or variations in CSF production/clearance rates[8–10]. Such inter-individual differences could lead to non-disease related differences in CSF protein levels[11], which could impact the overall performance of CSF biomarkers[12–14]. Hypothetically, adjustment for individual variability in CSF protein levels or to selected reference proteins could optimize the performance of already efficient CSF biomarkers, reduce false positive findings (by attenuating biomarker associations that are driven by the non-disease related variability), and increase the likelihood of making new biologically and clinically relevant discoveries. Similar approaches are commonly applied in other areas of medicine. Biomarkers in urine are for example normalized to a reference marker

[1]Department of Clinical Sciences in Malmö, Clinical Memory Research Unit, Lund University, Lund, Sweden. [2]Department of Clinical Sciences, Clinical Memory Research Unit, SciLifeLab, Lund University, Lund, Sweden. [3]Centre for Mathematical Sciences, Lund University, Lund, Sweden. [4]Department of Psychiatry and Neurochemistry, Institute of Neuroscience and Physiology, the Sahlgrenska Academy, University of Gothenburg, Mölndal, Sweden. [5]Clinical Neurochemistry Laboratory, Sahlgrenska University Hospital, Mölndal, Sweden. [6]Memory Clinic, Skåne University Hospital, Malmö, Sweden. ✉e-mail: linda.karlsson@med.lu.se; oskar.hansson@med.lu.se

(often creatinine) to adjust for variations in the urine concentration when the sample is collected at a single time point[15,16].

In AD research and clinical practice, CSF Aβ42 and P-tau, together with a CSF biomarker of neuronal injury (i.e., CSF total-tau or neurofilament light chain), can be used for AT(N) (amyloid, tau, neurodegeneration) in vivo classification of AD pathology[17,18]. This system makes it possible to categorize a person as biomarker positive or negative, where low CSF Aβ42 levels indicate Aβ plaque pathology ("A") and high CSF P-tau181 levels indicate tau tangle pathology ("T")[19–25]. AT(N) grouping is an effective way to differentiate individuals without AD (A-/T-) from those with AD (A+T+). However, many studies have reported findings in the group with isolated P-tau pathology (A-T+; i.e., both high Aβ42 and P-tau181), which is more controversial[17,26–28]. It is unclear if this A-T+ definition is biologically relevant or mainly a result of inter-individual differences in CSF levels (leading to more concentrated CSF in some individuals with higher levels of both Aβ42 and P-tau181).

Besides well-established CSF biomarkers used in clinical practice, CSF proteins are often studied to understand underlying disease mechanisms in humans affected by AD or other neurodegenerative diseases. In such studies, it has been suggested that CSF levels of many microglia-related proteins (like sTREM2 or TAM receptors [sAXL and sTYRO3]) are strongly correlated with CSF P-tau181 and increased not only in A+T+ individuals, but also in A-T+ individuals, linking these neuroinflammatory changes more to P-tau pathology than Aβ[20,21]. In addition, other CSF markers have been seen to correlate with CSF P-tau181 levels. For example, we and others have reported that the astrocytic biomarker YKL-40 and the Parkinson's disease-related biomarker α-synuclein are strongly associated with P-tau181 in CSF, which was interpreted as that these brain pathological changes co-vary[29–31]. It is unclear whether such findings are mainly driven by non-disease related inter-individual differences in CSF protein levels, or remain robust when accounting for this property. Moreover, the impact of CSF variability might also be of importance in proteomic studies, when identifying subpopulations with different CSF expression profiles[32], or in genome-wide protein quantitative trait loci (pQTL) studies, looking at associations between genetic variants and protein levels[14].

One striking example that highlights the potential of using a normalization protein in the context of AD CSF biomarkers exists. CSF Aβ42 shows improved concordance with amyloid positron emission tomography (PET, a well-established neuroimaging method to make aggregated brain amyloid in AD visible) when normalized for CSF Aβ40 levels, where the latter is not affected by the disease process[33,34]. Aβ40 is closely linked to Aβ42 since both peptides come from the same proteolytic pathway[35], but Aβ40 may also partly represent an individual's non-disease related CSF protein level and could potentially improve performance of other biomarkers as well. This idea has been tested for CSF P-tau181, where the results suggested that the diagnostic accuracy improved when adjusting for inter-individual differences in CSF Aβ40 levels[13]. In order to examine Aβ40's generalizability as a reference protein, it needs to be further evaluated. In addition to Aβ40, other efficient CSF reference proteins may exist that can improve the clinical performance of key CSF biomarkers.

Consequently, our overarching aim was to establish the concept of non-disease related inter-individual differences in CSF protein levels, and to search for optimal reference protein candidates that could be used to account for this variability in a robust way. This in order to more accurately detect disease-associated changes in key biomarkers. We analyzed 2944 CSF proteins (including CSF Aβ40) from 830 participants in a data-driven manner. We hypothesized that adjusting for certain reference proteins could improve the diagnostic accuracy of AD CSF biomarkers, and we evaluated this across a range of outcome measures, biomarkers, and AD cohorts. We also hypothesized that several previously reported CSF biomarker findings would be altered or attenuated when biomarkers were normalized to reference proteins. Specifically, we studied whether several strong and recognized correlations of CSF protein concentrations remained robust when normalizing to the identified reference proteins, both in relation to each other and to genetic variants.

## Results

The study included 830 participants from the Swedish BioFINDER-2 (BF2) cohort and 904 participants from Swedish BioFINDER-1 (BF1) cohort, all with complete OLINK CSF protein and CSF Aβ40 concentration measures (2,944 in BF2 and 369 in BF1). The included participants had either normal cognition (NC, $n = 263$ in BF2 and $n = 464$ in BF1), subjective cognitive decline (SCD, $n = 111$ in BF2 and $n = 172$ in BF1), mild cognitive impairment (MCI, $n = 193$ in BF2 and $n = 195$ in BF1), dementia ($n = 216$ in BF2 and $n = 73$ in BF1) or another neurodegenerative disease ($n = 47$, only in BF2). BF2 was randomly split into a training set (80%, $n=658$) and test set (20%, $n = 172$). Throughout this work, the training dataset of BF2 was used for all exploratory work. The BF2 test dataset was used to evaluate findings, and BF1 was used for external validation. To find and assess appropriate reference proteins, their performance was evaluated in three logistic regression models. The first model predicted tau-PET (a well-established neuroimaging method to demonstrate fibrillary tau deposition in AD) positivity with CSF P-tau181 (P-tau181→TauPET). The second predicted Aβ-PET (a well-established neuroimaging method to demonstrate fibrillary amyloid deposition in AD) positivity with CSF Aβ42 (Aβ42→AβPET). The third predicted future conversion to AD dementia with CSF P-tau181 (P-tau181→ADDconv). The first two models were used to search for suitable reference proteins while the third was used for validation of reference proteins. A flowchart and details of the complete reference search/evaluation pipeline, together with all data splitting details and demographics, are presented in Fig. 1 and Tab. 1. The main results focus on these analyses, with extended results in Supplementary Information.

### Many CSF proteins vary in concordance with the mean standardized CSF protein level

A mean standardized CSF protein level was calculated for each participant, representing how many standard deviations the concentration of CSF proteins in that participant deviated from the population mean (details in Methods: Statistical analyses, Eq. 1). We sorted the 2944 standardized CSF protein concentrations according to their associations with the mean standardized CSF protein level (Supplementary Fig. 1) and visualized the results in Fig. 2. Figure 2a indicates that, within a random subsample of individuals, there are several participants that systematically have high or low values across several hundred proteins. This phenomenon is evident across the full training dataset of 658 participants (Fig. 2b), where nearly half of all proteins measured appear to show highly consistent individual variation. The individual variability remained when stratifying on age and cognitive status (Supplementary Figs. 2, 3). When removing proteins of low detectability, this pattern becomes even clearer (Supplementary Fig. 4), emphasizing that most proteins that are highly expressed in CSF (and therefore likely to be nominated in CSF biomarker studies) vary in concordance with the mean standardized CSF protein level. The AD CSF biomarkers P-tau181 ($β = 0.34$, $P < 1e-20$) and Aβ42 ($β = 0.24$, $P < 1e-10$) were associated with the mean standardized CSF protein level (Fig. 2c). As expected, Aβ40 ($β = 0.44$, $P < 1e-37$) showed a stronger association with the mean standardized CSF protein level (Fig. 2c).

To further understand potential factors associated with mean standardized CSF protein level differences, we investigated the mean standardized CSF protein level's association with age, sex, education level, intracranial volume, gray matter volume and ventricular volume. In a multiple linear regression model, significant associations with a higher mean standardized CSF protein level were found for higher age ($β = 0.544$, $P = 4e-31$), male sex ($β = -0.159$, $P = 2e-4$), and lower ventricular volume ($β = -0.321$, $P = 5e-11$), see Supplementary Tab. 1.

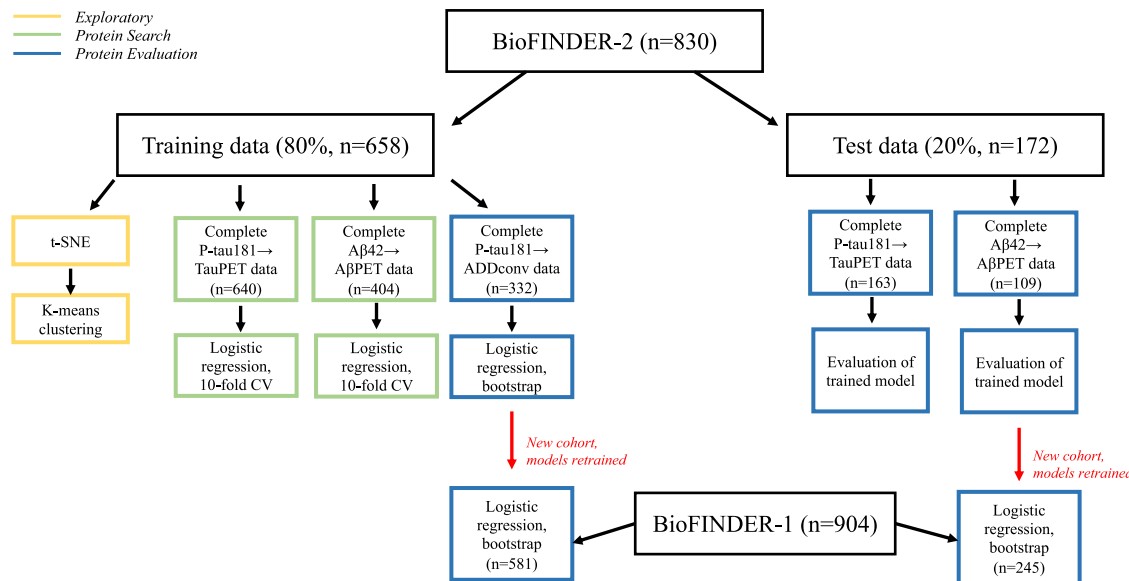

**Fig. 1 | Flowchart of reference protein search and evaluation pipeline.** BF2 was filtered by participants with CSF OLINK and CSF Aβ40 measurements (*n* = 830). Thereafter, the dataset was split into 80% training (*n* = 658) and 20% testing (*n* = 172). During the exploratory phase, all BF2 training data was used. Next, the two models P-tau181→TauPET and Aβ42→AβPET were used to search for reference proteins in the protein search phase. The proposed candidates were evaluated in corresponding models for unseen test data, and in a new third model P-tau181→ADDconv on the training data. The findings were further validated in the independent cohort BF1 (*n* = 904) for the two models Aβ42→AβPET and P-tau181→ADDconv. The model P-tau181→TauPET was not evaluated in BF1 as baseline tau-PET data did not exist. Complete data refers to no missing values for any of the relevant variables and was a filtering step in all models.

Similar results were seen when evaluating Aβ-negative cognitively normal participants only.

## A cluster with superior CSF reference protein qualities

We next examined clustering of the CSF protein concentrations to identify proteins of similar characteristics. We used t-SNE dimensionality reduction[36], applied to the high dimensional space of 658 participants (Fig. 3a). As the algorithm optimizes to preserve similarity of pairwise points, proteins of short distance in Fig. 3a can be interpreted as similarly expressed. In Fig. 3c–h, clustering characteristics of the t-SNE space are compared against several metrics relevant in search of reference proteins. A reference protein should be associated with the mean standardized CSF protein level (Fig. 3c). The mean standardized CSF protein level was highly associated with ventricular volume when adjusting for age and sex, suggesting that the mean standardized CSF level is partly driven by dilution. Therefore, lower levels of optimal reference proteins could be associated with larger ventricular volumes (Fig. 3d). A potential reference protein for a given model predictor, that can perform better than simply using the mean standardized CSF level of all proteins, should co-vary with the main predictor during normal physiology but not in disease. Therefore, an AD reference protein should likely have a high correlation with key biomarkers like P-tau181 and Aβ42 in cognitively unimpaired Aβ-negative participants, which are not considered to have the disease (Fig. 3e, f). Lastly, a well-performing reference protein should improve the predictive performance of key biomarkers, which was evaluated by comparing results of (i) using P-tau181 together with a potential reference protein to predict tau-PET outcome (P-tau181→Tau-PET) or (ii) using Aβ42 together with a potential reference protein to predict Aβ-PET outcome (Aβ42→AβPET) (Fig. 3g, h).

A semi-supervised K-means (*K* = 20) clustering algorithm[37] was utilized to divide the t-SNE space and identify a subset of potential reference proteins. The supervision was performed by selecting K and the random initialization so that clear structural clusters were separated and areas of overlap in Fig. 3c–h could be examined in more detail. The resulting K-means clustering can be seen in Fig. 3b. The clustering was relatively well in line with the OLINK panel division (see Supplementary Fig. 5), where the area of interest was likely to not benefit a certain panel.

The AUC of each cluster in the two regression models P-tau181→TauPET and Aβ42→AβPET were evaluated in Fig. 4a, b. The AUCs without using an individual reference were 0.865 and 0.934 respectively. By analyzing the performance cluster-wise, we aimed to target protein expression characteristics rather than single findings and hence remove top performances biased by data. As seen in both Fig. 4a, b, and as expected from the overlapping areas in Fig. 3, cluster 11 ($n_{proteins}$ = 219) stands out as the best performing cluster (mean AUC ± std: 0.896 ± 0.0187 and 0.947 ± 0.00698 for P-tau181→TauPET and Aβ42→AβPET respectively).

We performed cell type expression and cellular component pathway enrichment analyses on cluster 11 to further investigate the characteristics of promising reference proteins from a biological perspective. Cluster 11 had high expression in mainly neuronal cells (Supplementary Fig. 6) but showed no significant expression difference compared to the other 2725 OLINK proteins, which was assessed with a bootstrap enrichment test. Cluster 11 was enriched on cell surfaces and membranes (Supplementary Fig. 7). Additionally, we observed that the proteins of cluster 11 had higher brain expression than the other highly expressed OLINK proteins (Supplementary Tab. 2).

## Identification of general and biomarker-specific reference proteins

To further validate single robust reference proteins, specific candidates from cluster 11 that resulted in top AUC scores for the two models P-tau181→TauPET and Aβ42→AβPET were identified (Figs. 4c, d). While this extensive dataset of 2944 proteins allowed for great exploration possibilities, it also limited the validation opportunities in other cohorts. Additionally, some cohort-specific biases in our data were still expected, even after adding robustness by only looking at the subset of proteins from cluster 11. We hence did not expect small AUC differences between single proteins to be significant. Taking these factors into account, we selected three reference protein candidates in addition to Aβ40 (also in cluster 11), for further examination, based on the following selection criteria:

1. The protein was in cluster 11.

**Table. 1 | Demographics for all data paths in Fig. 1**

| | BioFINDER-2 (*n* = 830) | | | BioFINDER-1 (*n* = 904) |
|---|---|---|---|---|
| **Exploratory** | | | | |
| | BF2 train | | BF2 test | BF1 |
| *n* | 658 | | - | - |
| Age [years] | 68.2 (12.0) | | - | - |
| Sex male (%) | 351 (53.3%) | | - | - |
| Education [years] | 12.4 (3.75) | | - | - |
| MMSE | 26.3 (4.23) | | - | - |
| APOE ε4 carrier[a] | 319/655 | | - | - |
| NC/SCD/MCI/Dementia/Other | 196/94/147/176/45 | | - | - |

**P-tau181→TauPET (Predictors: CSF P-tau181, age, sex, individual reference, outcome: Tau-PET Braak I–IV > 1.36)**

| | BF2 train | BF2 train | BF2 test | BF2 test | BF1 | BF1 |
|---|---|---|---|---|---|---|
| | positive | negative | positive | negative | positive | negative |
| *n* | 170 | 470 | 32 | 131 | - | - |
| Age [years] | 72.3 (8.20) | 66.7 (12.7) | 72.9 (8.69) | 69.0 (10.7) | - | - |
| Sex male (%) | 85 (50%) | 249 (53%) | 14 (44%) | 68 (52%) | - | - |
| Education [years] | 12.8 (4.48) | 12.4 (3.51) | 12.0 (4.47) | 12.7 (3.57) | - | - |
| MMSE | 22.7 (5.05) | 27.6 (2.96) | 21.8 (5.29) | 27.3 (3.31) | - | - |
| APOE ε4 carrier[a] | 123/169 | 183/469 | 26/32 | 55/131 | - | - |
| NC/SCD/MCI/Dementia | 13/9/45/103 | 221/84/100/65 | 1/2/9/20 | 63/14/36/18 | | |
| CSF P-tau181 [pg/ml] | 37.5 (16.5) | 18.8 (8.09) | 37.1 (19.9) | 20.6 (13.3) | - | - |
| Tau-PET Braak I-IV [SUVR] | 2.08 (0.602) | 1.15 (0.0990) | 2.00 (0.562) | 1.16 (0.0920) | - | - |

**Aβ42→AβPET (Predictors: CSF Aβ42, age, sex, individual reference. Outcome: Amyloid-PET Centiloids > 20)**

| | BF2 train | BF2 train | BF2 test | BF2 test | BF1 | BF1 |
|---|---|---|---|---|---|---|
| | positive | negative | positive | negative | positive | negative |
| *n* | 133 | 272 | 40 | 71 | 101 | 144 |
| Age [years] | 71.5 (8.53) | 63.3 (14.5) | 72.9 (6.91) | 65.5 (11.8) | 72.5 (4.90) | 72.2 (5.82) |
| Sex male (%) | 66 (50%) | 138 (51%) | 16 (40%) | 38 (55%) | 54 (53%) | 69 (48%) |
| Education [years] | 12.9 (4.43) | 12.5 (3.43) | 12.1 (3.88) | 12.7 (3.04) | 11.3 (3.26) | 11.6 (3.35) |
| MMSE | 27.3 (2.35) | 28.5 (1.77) | 27.1 (2.26) | 28.6 (1.59) | 27.5 (1.63) | 28.5 (1.54) |
| APOE ε4 carrier[a] | 98/133 | 89/272 | 30/40 | 24/71 | 71/101 | 34/142 |
| NC/SCD/MCI/Dementia | 19/35/74/5 | 155/52/62/3 | 13/5/20/2 | 39/10/22/0 | 12/29/60/0 | 61/39/44/0 |
| CSF Aβ42 [pg/ml] | 972 (275) | 1960 (737) | 953 (300) | 2030 (760) | 743 (292) | 1586 (625) |
| Amyloid-PET [Centiloids] | 77.8 (32.1) | −6.12 (7.64) | 66.6 (30.9) | −6.59 (7.42) | 82.5 (33.6) | 2.41 (8.33) |

**P-tau181→ADDconv (Predictors: CSF P-tau181, age, sex, individual reference. Outcome: Conversion to AD dementia[b])**

| | BF2 train | BF2 train | BF2 test | BF2 test | BF1 | BF1 |
|---|---|---|---|---|---|---|
| | positive | negative | positive | negative | positive | negative |
| *n* | 40 | 292 | 9 | 75 | 145 | 436 |
| Age [years] | 71.7 (8.32) | 63.6 (14.5) | 73.4 (6.78) | 66.2 (11.4) | 72.8 (4.80) | 71.8 (5.65) |
| Sex male (%) | 14 (40%) | 148 (51%) | 5 (56%) | 34 (45%) | 75 (52%) | 182 (42%) |
| Education [years] | 14.1 (5.69) | 12.5 (3.40) | 12.4 (3.05) | 12.7 (3.18) | 11.4 (3.23) | 12.2 (3.57) |
| MMSE | 26.8 (1.85) | 28.8 (1.41) | 26.2 (1.86) | 29.0 (1.27) | 27.1 (1.73) | 28.9 (1.21) |
| APOE ε4 carrier[a] | 34/39 | 74/198 | 7/9 | 32/75 | 106/145 | 125/434 |
| CSF P-tau181 [pg/ml] | 36.8 (13.6) | 19.1 (8.34) | 33.8 (6.61) | 18.0 (9.33) | 35.4 (15.2) | 19.2 (7.44) |
| NC/SCD/MCI | 1/2/37 | 176/93/23 | 0/1/8 | 52/15/8 | 6/35/104 | 263/124/49 |
| Conversion time [years] | 1.88 (1.13) | - | 1.62 (1.05) | - | 3.31 (2.06) | - |

Details of all data used for exploration, protein search and protein evaluation in concordance with the different data paths in Fig. 1. The regression models all include age, sex and individual reference, but differ by main predictor and outcome. The differences generated a variation in number of participants and demographics for each model, depending on the data available. Exploratory work was only performed on training data in BF2. In BF1, tau PET data did not exist. mini mental state examination (*MMSE*), positron emission tomography (*PET*), normal cognition (*NC*), subjective cognitive decline (*SCD*), mild cognitive impairment (*MCI*), standardized uptake value ratio (SUVR).
[a]missing data for some individuals.
[b]negative = stable cognition for at least 2 years.

2. The inclusion of the protein led to an increased AUC score for both the P-tau181→TauPET and Aβ42→AβPET model.
3. The protein was measured in our independent validation cohort (BF1).

The resulting three proteins, NTRK3, NTRK2 and BLMH, are marked out in Fig. 4c. See Supplementary Note 1: Reference Protein Candidate Profiles for a biological description of the three proteins and BF2 data information further confirming the proteins' potential as

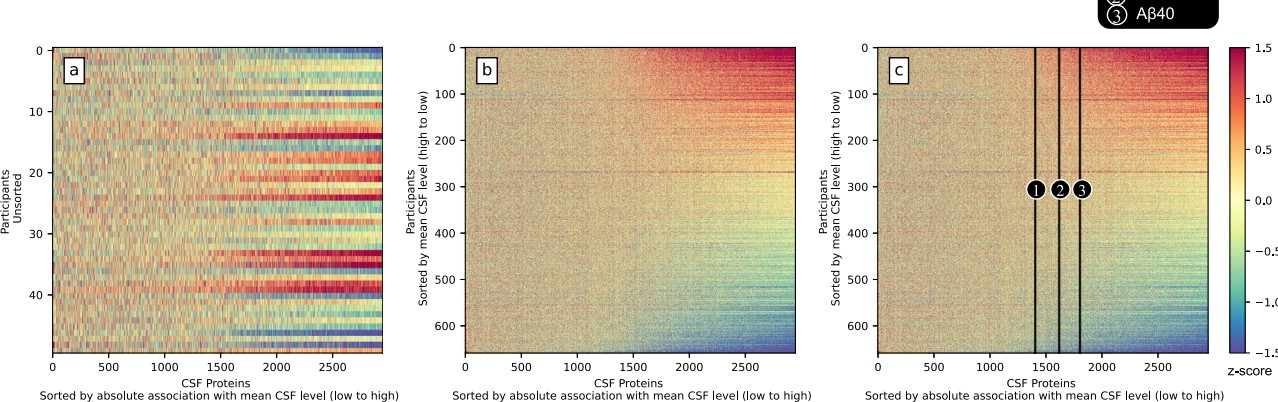

**Fig. 2 | Many CSF proteins vary in concordance with an individual protein level.** For each participant (row), the standardized concentration of 2944 CSF proteins, sorted by increasing absolute association with the mean standardized CSF protein level (Supplementary Fig. 1), is displayed. Systematic blue horizontal lines can be seen for individuals with consistently low values across most proteins, and correspondingly red horizontal lines for individuals with high values across most proteins (all relative to the total sample). **a** a subset of 50 randomly selected participants. **b** all 658 participants sorted by mean standardized CSF level. **c** same as (**b**) but also including biomarkers Aβ42 and P-tau181, which together with Aβ40 are marked out. The further right the protein is located, the more associated with the mean standardized CSF level and therefore more strongly co-varying with the mean standardized CSF protein level.

suitable references (e.g., high association with mean standardized CSF level, small concentration differences between diagnostic groups, high association with the main predictors, low model performance when used without a main predictor and high detectability in majority of subjects [missing frequency ≤ 0.0008]).

We hypothesized that each main predictor may have one or several optimal reference candidates. An optimal reference is co-varying to a higher extent with the main predictor during normal physiology but not in disease. For Aβ42, a natural biomarker specific reference is Aβ40, as both are generated from the amyloid precursor protein (APP)[35] and hence largely follow the same biological pathway. To further evaluate this concept of biomarker-specific reference proteins, we investigated the possibility of finding exceptionally high-performing references for CSF P-tau181. There is high relevance in identifying optimal references for CSF P-tau181, as there is no state-of-the-art way of normalizing this biomarker. Furthermore, as CSF P-tau181 was more correlated with the mean standardized CSF level than Aβ42, it may have more potential for improvement when adjusting for a reference protein. This idea was strengthened by the results shown in Fig. 4, where the AUC improvement was considerably larger for P-tau181→TauPET than Aβ42→AβPET when adjusting for a reference (max AUC improvement: 0.076 versus 0.031, cluster 11 mean AUC improvement: 0.031 versus 0.013). We therefore identified the three proteins in cluster 11 that improved P-tau181→TauPET the most: CBLN4, PTPRN2 and PTPRS in Fig. 4d. See Supplementary Note 1: Reference Protein Candidate Profiles for a biological description of the three proteins and BF2 data information further confirming the proteins' potential as suitable references.

**Adjusting for reference proteins improves biomarker performance**

In accordance with the flowchart in Fig. 1, five combinations of models and datasets were evaluated. For all models, performance was compared between no reference and Aβ40, mean standardized CSF level and the three general reference protein candidates NTRK3, NTRK2, BLMH as reference. Additionally, as the three candidates were highly correlated (Pearson correlation 0.85–0.91, see Supplementary Fig. 8a), the first component of a singular value decomposition was also evaluated as a possible reference, created from the three candidates (SVD1). For all models using BF2 data, the P-tau181-specific reference protein candidates (CBLN4, PTPRN2 and PTPRS, Pearson correlation 0.79–0.89, see Supplementary Fig. 8b) and their corresponding first

component of a singular value decomposition (SVD2) were evaluated as well.

The two models P-tau181→TauPET and Aβ42→AβPET were retrained on the full BF2 training dataset and evaluated on the BF2 test dataset (Fig. 5a–d and Supplementary Tab. 3). For both models and all references, the performance significantly increased (AUC = 0.895–0.946 and 0.977–0.992, $P < 0.05$, for P-tau181→TauPET and Aβ42→AβPET respectively) compared to no reference (AUCs = 0.828 and 0.966). Additionally, using a single reference protein or an SVD of three candidates (AUCs = 0.908–0.946 and 0.982–0.992) also outperformed using the mean standardized CSF level as reference (AUCs = 0.895 and 0.977). For P-tau181→TauPET, top performance was reached when adjusting for SVD2 (AUC = 0.946, $P < 0.01$), closely followed by CBLN4 (AUC = 0.944, $P < 0.01$). For Aβ42→AβPET, top performance was reached when adjusting for Aβ40 (AUC = 0.992, $P < 0.05$).

To validate the generalizability of the reference candidates, Aβ42→AβPET was applied in BF1 and the new model P-tau181→ADDconv was applied in both BF2 and BF1 (Fig. 5e–g and Supplementary Table. 3). Note that measurements of CBLN4, PTPRN2 and PTPRS were not available in BF1. Again, using no reference consistently resulted in the lowest performance (AUCs = 0.866, 0.880 and 0.916 for P-tau181→ADDconv in BF2, P-tau181→ADDconv in BF1, and Aβ42→AβPET in BF1 respectively). For P-tau181→ADDconv in BF2 (Fig. 5e), no significant improvements were achieved, most likely due to the small sample size ($n_{pos} = 40$, $n_{neg} = 292$). However, the same trends as in Fig. 5c were seen, where the proposed P-tau181-specific reference candidates again achieved top performance, together with corresponding SVD2 (AUCs = 0.922–0.930). For the same model P-tau181→ADDconv in BF1 (Fig. 5f), the available data was larger ($n_{pos} = 145$, $n_{neg} = 436$), and several significant improvements were achieved both compared to no reference and mean standardized CSF level as reference, with NTRK3 showing best performance (AUC = 0.935, $P < 0.01$). For Aβ42→AβPET in BF1 (Fig. 5g), a significant improvement was only achieved with Aβ40 (AUC = 0.970, $P < 0.05$), which was clearly superior to all other tested references.

**Reference proteins explain discordance between CSF and PET tau positivity**

We investigated how adjusting for an individual reference protein affected concordance between AT(N) grouping for CSF P-tau181 and tau-PET (Fig. 6). For this analysis, all participants from the BF2

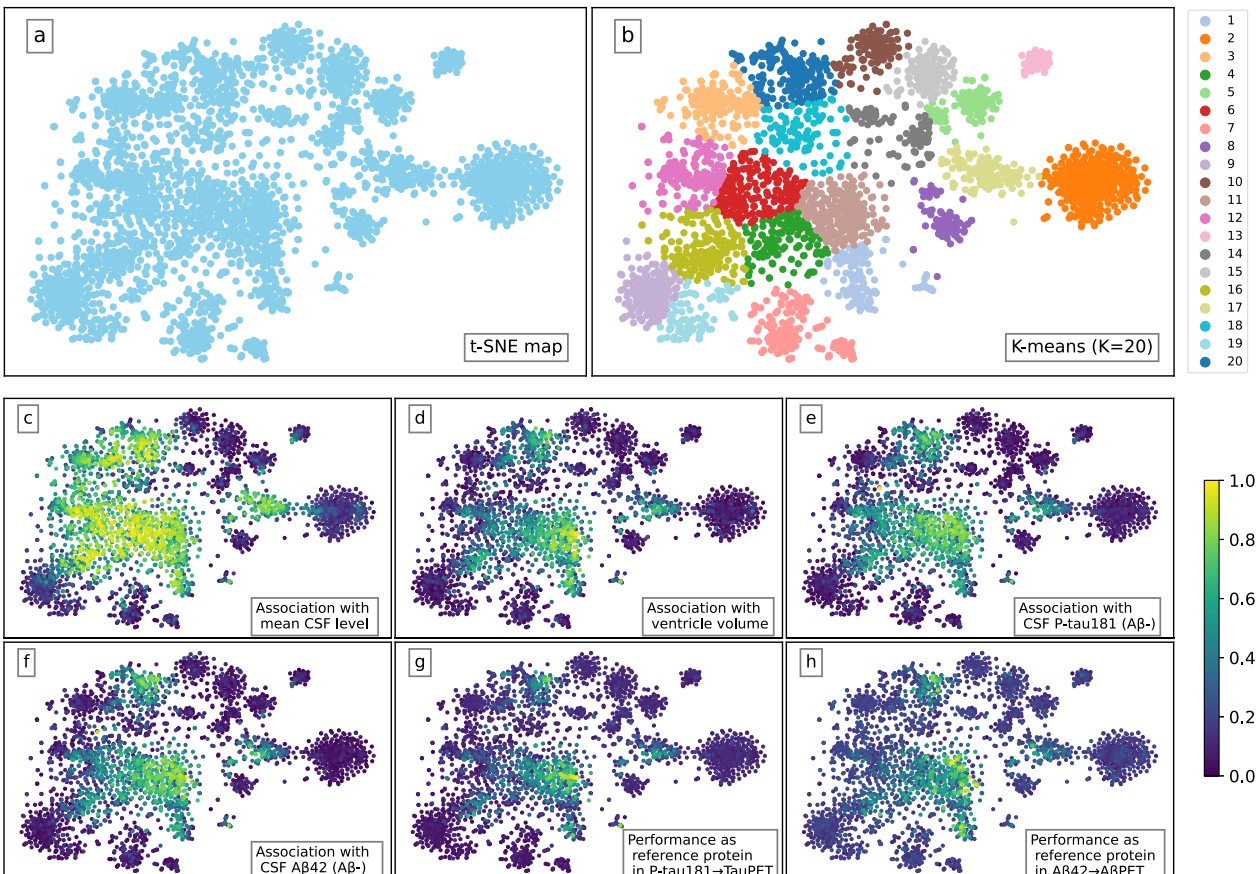

**Fig. 3 | Dimensionality reduction reveals a cluster of CSF proteins with desired reference protein characteristics.** T-distributed stochastic neighbor embedding (t-SNE), reducing the high dimensional space of 658 participants to a two-dimensional one. Each scatter point illustrates one of the 2944 CSF proteins, with relative similarity of pairwise proteins aimed to be preserved. In (**c–h**), min-max scaling for six different criteria has been performed to visualize relative differences within the space, all plots ranging between 0–1 (dark blue to yellow). In (**c–f**) associations are analyzed as absolute β-coefficients, all adjusted for age and sex. **a** The raw t-SNE map. **b** Semi-supervised K-means (K = 20) clustering of t-SNE map, aiming to separate the evident clusters from t-SNE dimensionality reduction and areas of overlap in (**c–h**). t-SNE map colored by (**c**) absolute association with mean standardized CSF level; (**d**) absolute association with ventricle volume; (**e**) absolute association with biomarker CSF P-tau181 (cognitively unimpaired Aβ-negative participants only); (**f**) absolute association with biomarker CSF Aβ42 (cognitively unimpaired Aβ-negative participants only); (**g**) model performance when used as reference protein in P-tau181→TauPET; (**h**) model performance when used as reference protein in Aβ42→AβPET. Source data are provided as a Source Data file.

training dataset with tau-PET data were included (n = 640). A-grouping was performed with CSF Aβ42/Aβ40 (cutoff 0.08[38]). T-grouping was made using (1) no reference (Fig. 6a, CSF cutoff of P-tau181 > 21.8 pg/ml, as in e.g., ref. 20) and (2) the reference protein candidate CBLN4 (Fig. 6c, CSF cutoff of P-tau181 > 39.0 + 10.1$c_{CBLN4}$, adapted from a logistic regression model). In addition, the grouping methods were compared against (3) tau-PET grouping (Fig. 6b, d). The concordance between CSF and PET grouping increased both visually and quantitively when not requiring a cutoff based on CSF P-tau181 only, but also accounting for the reference protein. The accuracy increased from 76% to 89% (Fig. 6e). Particularly notable is that the A-T+ group was reduced from n = 37 to n = 10 when using a reference protein. Among the ten CSF A-T+, most were close to the decision boundary of being A-T- and only one was classified as A-T+ by PET, indicating that this group could be even further reduced.

Examples of how previously published research results of P-tau181[20,21] were affected by this CSF AT(N) grouping improvement can be seen in Fig. 7a and Supplementary Results: Adjusting for a Reference in P-tau181 Applications. Figure 7a and Supplementary Figs. 9–11 show how concentrations of sTREM2, sAXL and sTyro3 turned out to be substantially less differentiable between AT(N) groups when adjusting for a reference protein during grouping. These latter results were clearly more similar to results obtained when using PET (instead of CSF) to define AT(N) groups, indicating that the reported relations between AT(N) and these microglia-related proteins were strongly driven by inter-individual variability in CSF protein levels. Further examples of this effect can be seen in Supplementary Tables 4, 5, where correlations between P-tau181 and sTREM2, sAXL, sTyro3 and α-synuclein were clearly attenuated when adjusting for a reference protein.

### Reference proteins often attenuate CSF biomarker associations

Examples of how other CSF proteins are affected by adjusting for a reference protein are seen in Fig. 7b, c and Supplementary Results: Change in Results when Adjusting for Reference Proteins. In Fig. 7b and Supplementary Fig. 12, partial correlations between ten established biomarkers from the NeuroToolKit assay panel proteins with and without adjusting for a reference are given. In general, correlations decreased when adjusting for a reference. This was seen most evidently for the cognitively unimpaired Aβ-negative participants and for proteins highly associated with the mean standardized CSF level, such as CSF levels of sTREM2, YKL-40 and tau. Additionally, examples of reference proteins' impact on associations between certain CSF proteins and genetic variants are presented. This includes strengthened

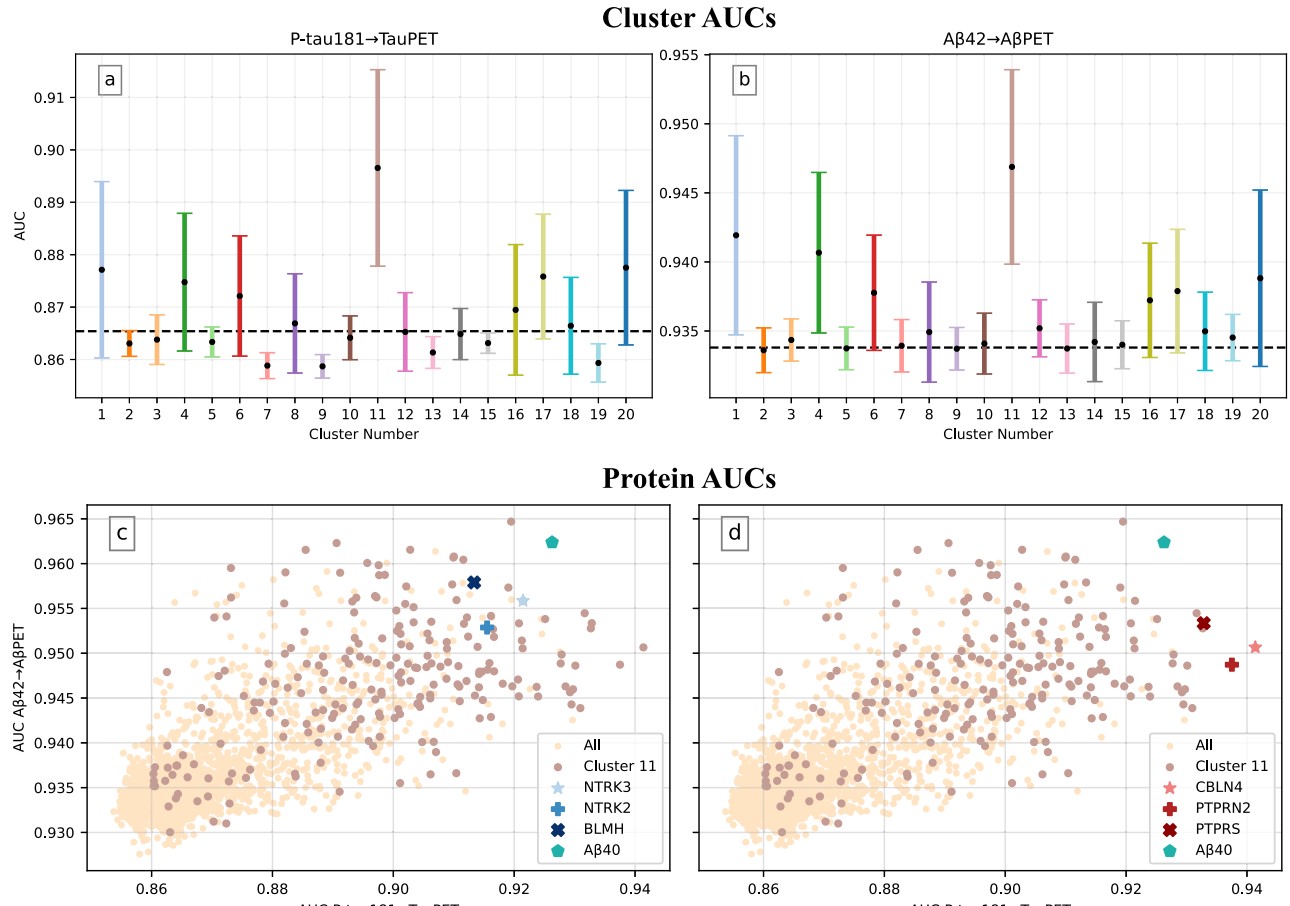

**Cluster AUCs**

**Protein AUCs**

**Fig. 4 | Cluster 11 stands out as superior when used as reference in models P-tau181→TauPET and Aβ42→AβPET.** In (**a**, **b**) each error bar represents the cluster's mean performance ± one standard deviation when the proteins of the cluster are used as a participant's individual reference (one protein at a time, all proteins evaluated once, adjusting for age and sex). The dashed lines correspond to the models' AUCs without using a reference (0.865 in **a**) and (0.934 in **b**). Each cluster is colored as in Fig. 3b. For both models, cluster 11 stands out as the best performing cluster on average. In (**c**, **d**) every scatter point corresponds to the result when evaluating that protein as individual reference. Consequently, the top right corner contains proteins of most interest. Proteins in cluster 11, Aβ40 and three other

reference protein candidates are highlighted. Note that (**c**, **d**) are identical apart from the highlighted markers. **a** results for P-tau181→TauPET by cluster. **b** results for Aβ42→AβPET by cluster. **c** proteins NTRK3, NTRK2 and BLMH selected as general reference proteins, after enforcing the selection criteria. **d** proteins CBLN4, PTPRN2 and PTPRS selected as P-tau181 specific reference proteins, as they improved P-tau181→TauPET the most. All models were evaluated using 10-fold-cross-validation on the BF2 training dataset. Number of proteins in each cluster: ($n_1$= 135, $n_2$= 305, $n_3$= 183, $n_4$= 132, $n_5$= 107, $n_6$= 206, $n_7$= 163, $n_8$= 79, $n_9$= 171, $n_{10}$= 109, $n_{11}$= 219, $n_{12}$= 165, $n_{13}$= 53, $n_{14}$= 103, $n_{15}$= 119, $n_{16}$= 167, $n_{17}$= 133, $n_{18}$= 112, $n_{19}$= 87, $n_{20}$= 196). Source data are provided as a Source Data file.

associations of apolipoprotein E (APOE) ε4 alleles with protein levels of ApoE4 and reduced/disappeared associations of trans-protein quantitative trait loci (pQTL) with genes from the GMNC-OSTN region (previously shown to be associated with variations in ventricular volume and suggested to be linked to both CSF P-tau and several other CSF proteins[14,39]), Fig. 7c, Supplementary Tables 6–8 and Supplementary Fig. 13.

## Discussion

We establish the existence of non-disease related individual CSF variability (Fig. 2), which explain a considerable part of variation in CSF biomarkers. We conclude that many proteins are affected by individual CSF levels, including proteins relevant in AD, and may benefit from adjustment of this factor when used as biomarkers. We identify a robust subset of potential reference proteins ("cluster 11") from which we further characterize six specific reference protein candidates (NTRK3, NTRK2, BLMH, CBLN4, PTPRN2 and PTPRS) that can significantly improve the accuracy of key AD biomarkers (Figs. 3, 4). The results are validated on unseen test data and in an independent cohort (Fig. 5). We provide evidence that Aβ40 works well as a reference protein, not only for Aβ42, but for P-tau181 and other CSF biomarkers

as well. Further, we show that several previously reported CSF biomarker classifications and associations increased in accuracy/effect or were greatly diminished when adjusting for reference proteins (Figs. 5–7). This implies that future studies should account for a reference protein to improve biomarker sensitivity and/or ensure that observed CSF biomarker relationships are not mainly driven by non-disease related differences in CSF protein levels. Our work focuses on biomarkers in AD, but since the issue of CSF variability is not AD specific, this concept likely has broad relevance across all neurological and psychiatric conditions where CSF biomarkers are used.

In general, when using disease-related biomarkers, we observe an altered level of a protein in a group of patients and relate it to a disease. Previously, the definition of "an altered CSF level" has been that the level is different from what is normal on a group level (for example when using a universal cutoff/threshold). Since we see inter-individual differences of the mean standardized CSF protein levels not related to disease (Fig. 2 and Supplementary Fig. 3), this means that false positives (individuals with high biomarker levels not due to disease, but just because many protein concentrations in the CSF are relatively high) and false negatives (individuals with low biomarker levels but have the disease, but just because many protein concentrations in the

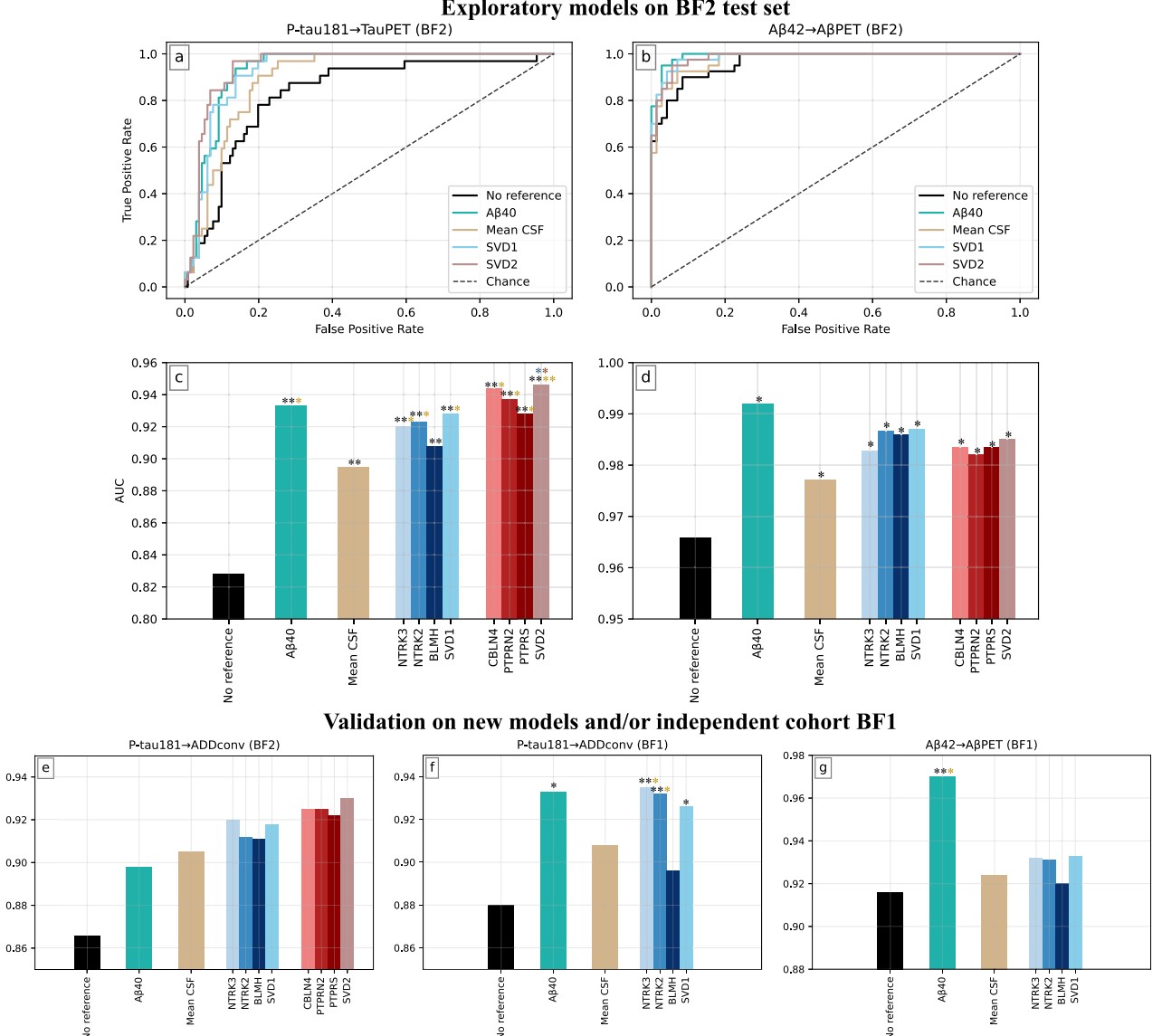

**Fig. 5 | Performance evaluation of all models with and without references.** ROC curves in (**a**, **b**), and corresponding AUCs in (**c**, **d**), for the two models P-tau181→TauPET and Aβ42→AβPET, evaluated on the BF2 test dataset. AUCs for (**e**) model P-tau181→ADDconv on the training dataset BF2, (**f**) model P-tau181→ADDconv on BF1 and (**g**) model Aβ42→AβPET on BF1. No reference corresponds to the model without use of a reference protein, consistently outperformed by all tested references: Aβ40, mean standardized CSF level, NTRK3, NTRK2, BLMH, SVD1, CBLN4, PTPRN2, PTPRS and SVD2. Quantitative details, including exact *P* values, can be found in Supplementary Table. 3. For visualization purposes, the ROC curves do not show all protein candidates but solely the corresponding SVDs. AUCs were compared with one-sided ROC test using bootstrapping ($n_{iter}$ = 2000). *P* values were adjusted for multiple comparisons by Benjamini–Hochberg method. All models were adjusted for age and sex. **P* < 0.05, ***P* < 0.01 compared to the bar of same color as asterisk.

CSF are relatively low) can occur from this commonly used method. Proteins that are majorly changed in CSF in a brain-disease like AD can clearly be identified even without a reference protein (like using CSF Aβ42 without Ab40 as a reference). But when normalizing to reference proteins, we can likely detect more proteins that are changed in a brain disease and further avoid some false positive findings (as illustrated by many examples in this manuscript).

The existence of individual variability in CSF protein levels provides valuable insights on a CSF characteristic that must be acknowledged and further explored by the field of CSF biomarkers. When investigating potential biological factors associated with the mean standardized CSF protein level differences, we found that an increased mean standardized CSF protein level was seen in males and was associated with higher age. This may be connected to other sex and age-related CSF dynamics, like differences in CSF pressure

and CSF production and clearance rates. These dynamical differences have previously been observed in both human and animal studies[40–43]. A reduced CSF production and clearance rate, as seen during aging, may for example contribute to longer accumulation time of CSF proteins, resulting in an increased individual CSF protein level. Additionally, as the mean standardized CSF protein level was associated with ventricular volume (but not intracranial volume) when adjusting for age and sex, we believe that CSF dilution is an important factor explaining why these individual differences exist. As CSF fills the ventricles, it is reasonable that the size of the ventricles affects the produced CSF volume independently of CSF protein secretion, leading to these CSF dilution differences. Supporting this hypothesis is that individuals with idiopathic normal pressure hydrocephalus (iNPH), characterized by an abnormal buildup of CSF resulting in enlarged ventricles, also show

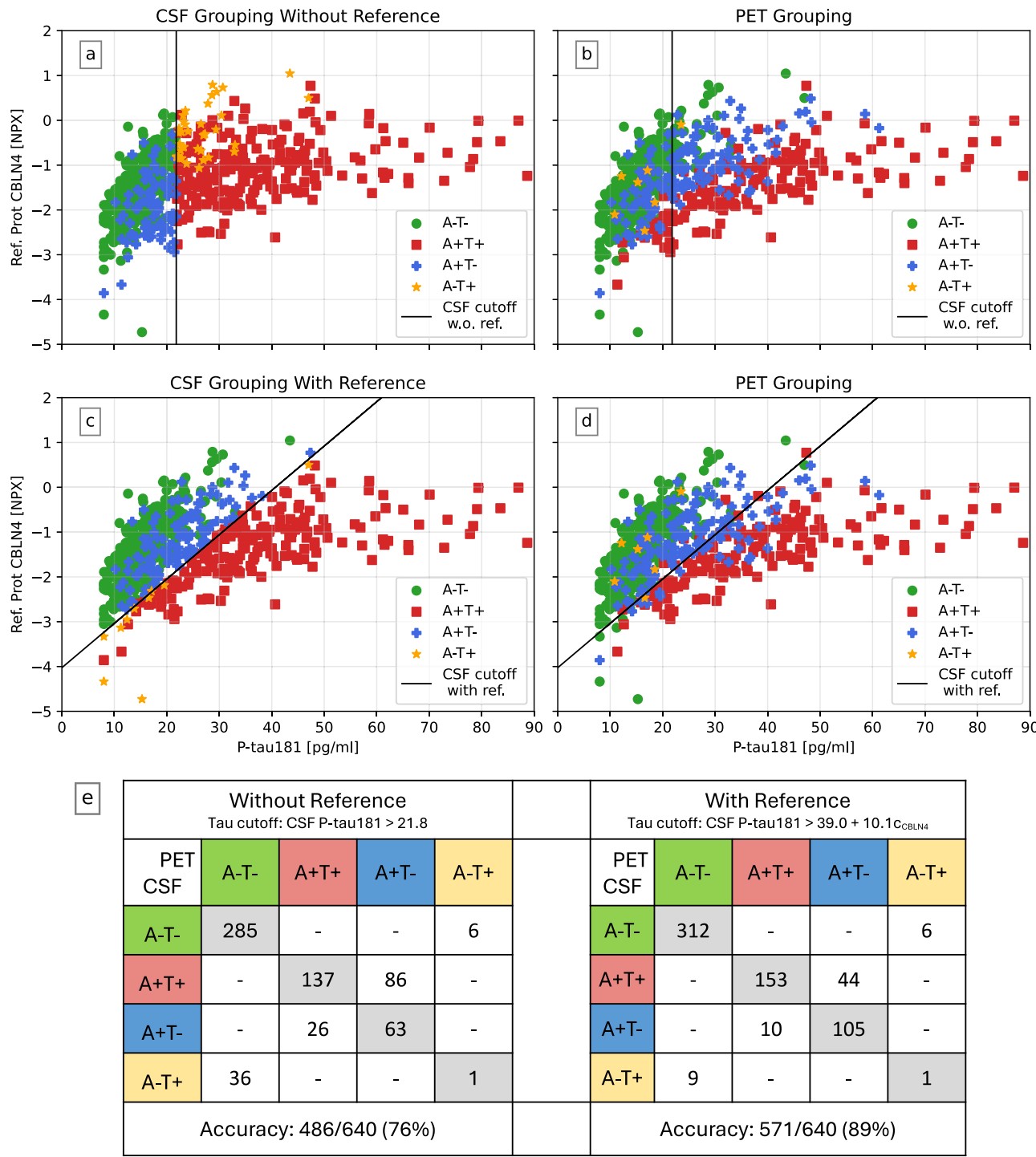

**Fig. 6 | Adjusting for an individual reference protein (here CBLN4) results in a better AT(N) grouping concordance between CSF P-tau181 and tau-PET.** In (**a**–**d**), the x-axis is a participant's CSF P-tau181 concentrations, and the y-axis the suggested reference protein CBLN4. In (**a**, **c**), CSF P-tau181 and CSF Aβ42/Aβ40 (cutoff 0.08) has been used to group participants. In (**b**, **d**), a tau-PET composite corresponding to Braak I-IV with ROIs > 1.36 and CSF Aβ42/Aβ40 was used to group participants. To create a cutoff for CSF P-tau181 (black line), the reference protein CBLN4 was adjusted for in (**c**, **d**) (cutoff: CSF P-tau181 > 39.0 + 10.1$c_{CBLN4}$) but not in (**a**, **b**) (cutoff: CSF P-tau181 > 21.8). The concordance between CSF P-tau181 and tau-PET grouping increased when not requiring a vertical cutoff line but allowing for it to have a slope. In (**e**) corresponding concordance matrices of PET and CSF with and without using a reference for CSF P-tau181 can be seen. The concordance increased from 76% to 89% when adjusting for the reference protein. Particularly notable is that the A-T+ group (which is pathophysiologically difficult to explain) was reduced from *n* = 37 to *n* = 10 when using a reference protein, again in higher concordance with grouping with PET.

substantially low (diluted) CSF AD biomarker levels compared to healthy subjects[44]. We acknowledge that the hypothesis of CSF dynamics being a driving factor of inter-individual CSF variability needs to be confirmed in future studies before concluded, by for

example examining relationships between CSF protein levels and CSF clearance and production rates[45].

The results of our study challenge previously held notions of strong relationships between several CSF biomarkers in AD. We show

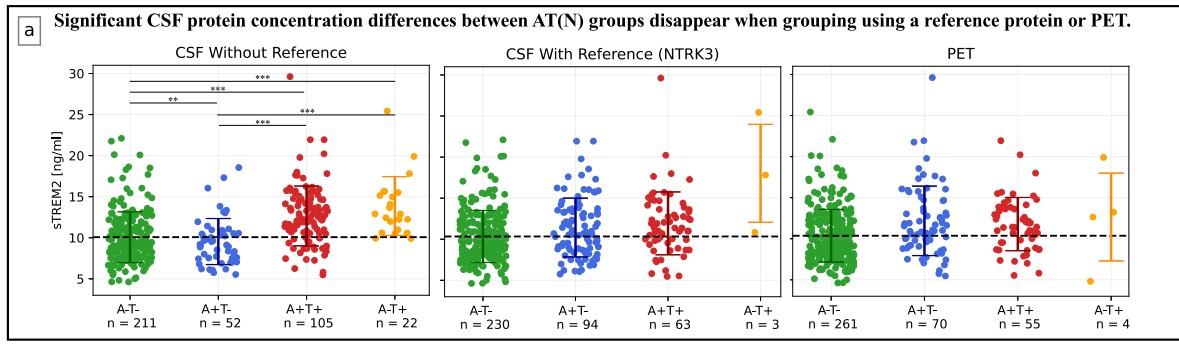

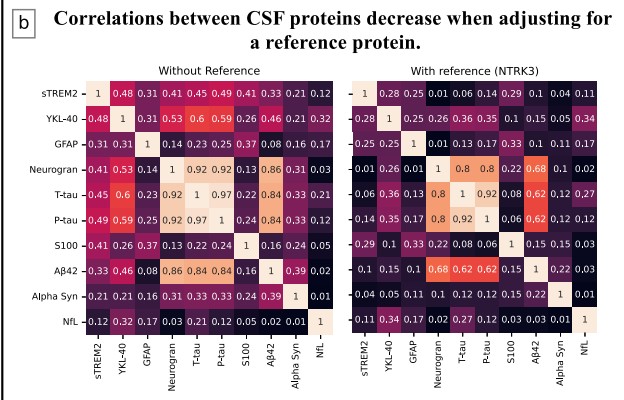

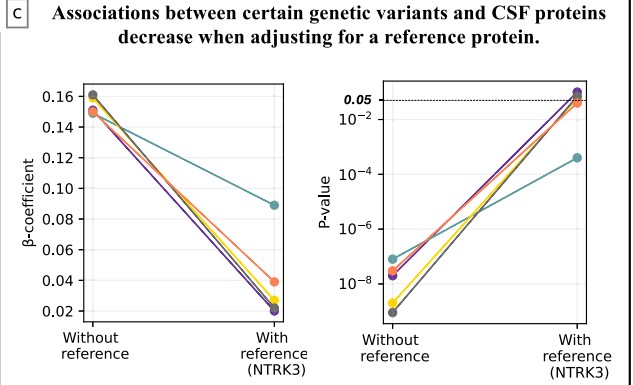

**Fig. 7 | Several CSF proteins are affected by adjusting for a reference protein.** Further details of these results and extensive analyses on similar findings can be found in Supplementary Information. **a** During AT(N) grouping assessed by CSF, adjusting for a reference protein created a better concordance with PET when comparing concentration differences of CSF sTREM2. The significant findings from not using a reference protein during grouping most likely appeared due to mean standardized CSF protein level differences between groups. This analysis included NC, SCD and MCI BF2 participants. Error bars represent the mean concentration ± one standard deviation. *P* values (adjusted for multiple comparisons) were assessed by a one-way ANCOVA adjusted for age and sex. Source data are provided as a Source Data file. **b** Partial correlation matrices for ten NeuroToolKit proteins in BF2 cognitively unimpaired Aβ-negative participants with and without adjusting for

a reference protein, always adjusting for age and sex. Proteins were sorted according to decreasing association with mean standardized CSF level (see Supplementary Table 10). Almost all correlations were severely reduced when adjusting for a reference protein, most clearly seen for proteins highly associated with the mean standardized CSF level (top rows). **c** Results from protein quantitative trait loci (pQTL) analyses where associations between certain CSF proteins and genetic variants have been identified. Several CSF trans-pQTL associations of the GMNC-OSTN region showed severely weakened relationships when adjusting for a reference protein. For details, see Supplementary Results: CSF pQTL Analysis and Supplementary Table 8. These models included BF1 participants (*n*=1445) and were adjusted for age, sex, dementia diagnosis and ten genetic principal components.

that many of these correlations were mainly driven by individual variability in CSF protein levels, as they did not remain robust when accounting for a reference protein. Associations were markedly reduced for P-tau181 versus microglia-related proteins (e.g., sTREM2 or TAM receptors [sAXL and sTYRO3]), astrocytic biomarker YKL-40 and the Parkinson's disease-related biomarker α-synuclein, indicating that these biological processes/pathologies may not be as related to tau-pathology as both we and others have previously suggested[20,21,29–31].

Our findings provide important insight of how biased inter-correlations between CSF biomarkers and biomarker groups can appear when not accounting for non-disease related differences in CSF protein levels. Specifically highlighting this was how the use of a reference protein for P-tau181 in the AT(N) grouping context (Fig. 6e) showed that the A-T+ group (which is pathophysiologically difficult to explain[17]) was reduced from *n* = 37 to *n* = 10 when using a reference protein. This result is more in line with studies using PET to classify individuals according to the AT(N) system[46]. The characteristics of A-T+ is highly researched and discussed[25–28,47–49]. We show that this group largely consists of individuals with high overall mean standardized CSF protein levels (rather than any specific disease marker), which may explain why many A-T+ individuals have high CSF concentrations of other proteins than P-tau181[20,21,27]. Other conclusions from results in the CSF-based A-T+ group, like hypotheses about tauopathy (T+) not affecting cognition[19], can be highly influenced by erroneous classification of individuals with high overall CSF levels into an A-T+ group.

Throughout the evaluation, three general reference proteins (NTRK3, NTRK2 and BLMH) and Aβ40 were examined. For all P-tau181 associations, NTRK3, NTRK2 and Aβ40 performed similarly, while BLMH performed slightly inferior. As reference to Aβ42, Aβ40 was superior. Aβ40 consistently provided improved accuracy and top performance when used as reference for P-tau181 as well, congruent with ref. 13. As Aβ40 is already often measured in CSF studies, the performance of Aβ40 as a reference can easily be further validated for other biomarkers and cohorts. Aβ40 could be a suitable first individual reference to adjust for when working with CSF AD biomarkers.

We hypothesized that each CSF biomarker might have one or several optimal reference proteins that co-vary highly with the main pre-dictor during normal physiology but not in disease, in addition to representing an individual CSF protein level. This idea is strengthened by the fact that the top reference protein candidates showed high corre-lation with the main predictors in cognitively unimpaired Aβ-negative participants but were not predictive of the outcome when applied in models alone (AUCs between 0.62 and 0.67, as seen in Supplementary Note 1: Reference Protein Candidate Profiles). Additionally, simply adjusting for the mean standardized CSF level never resulted in best performance, which also is in line with reference markers in other areas of medicine. As a comparison, one can consider how biomarkers in urine are processed today, where the normalization method of urine is critical due to dilution differences. For urine biomarkers, total protein con-centration can be used as a standardization technique, but it is

considered less precise compared to using ratios to other excreted small molecules or proteins like creatinine[15,16]. This concept is very similar to what we propose with CSF reference proteins in this work.

For Aβ42, a reference that will outperform Aβ40 is unlikely to emerge, as this peptide so closely follows the same biological pathway as Aβ42. Previous work has shown the benefit of adjusting Aβ42 levels with Aβ40[33,34], and in extension, our results confirm that Aβ40 performed best as reference for Aβ42 when compared to other possible CSF references. For P-tau181, no such optimal reference has previously been suggested, but three possible candidates (CBLN4, PTPRN2 and PTPRS) were evaluated in this paper. The candidates performed best for two P-tau181 models, but these candidates were only evaluated in BF2 and did not significantly outperform other suggested candidates. Additionally, none of them are as obviously associated with P-tau181 in regards to biological pathway as Aβ40 is with Aβ42, which further decreases their probability of being optimal P-tau181 specific references. However, other factors could make CSF protein concentrations co-vary, such as cellular localization as well as protein size, charge, and solubility. The properties and generalizability of these candidates should be further examined before they can be claimed optimal or non-optimal reference proteins for P-tau181.

As seen in Supplementary Note 1: Reference Protein Candidate Profiles, the six reference protein candidates (NTRK3, NTRK2, BLMH, CBLN4, PTPRN2 and PTPRS) all had higher association with the mean standardized CSF protein level (partial correlations 0.62–0.80) than Aβ40 (partial correlation 0.48). Many of them were cell surface receptors and involved in cell survival and differentiation. Most had enhanced brain specificity, but low regional brain specificity. From their performance as reference proteins in BF2, we cannot conclude that these specific six candidates were significantly superior reference proteins. On the contrary, several proteins shared similar expression characteristics that were beneficial for a reference protein. We therefore presented a subset of many such proteins by using a data-driven approach to group them into the cluster referred to as cluster 11. From the cell type expression analysis, we found that cluster 11 was representative of the entire set of 2944 proteins with majority of proteins highly expressed in neuronal cells. The proteins of cluster 11 were enriched on cell surfaces and membranes and are therefore probably constantly shedded into the CSF during normal physiology, which can explain why they maintain a relatively consistent concentration level representative of the individual CSF protein level. Additionally, as a large proportion of the total CSF proteome consists of blood-derived proteins (~80%; mainly albumin[11]), many CSF proteins are likely affected by the integrity of the blood-CSF barrier. As the proteins of cluster 11 showed higher brain expression than the other highly expressed CSF proteins, they are not as impacted by blood-CSF barrier permeability and therefore likely more robust reference proteins for biomarkers in brain diseases.

A potential limitation of the study was that we only had the possibility to validate parts of the results in an independent cohort. While being a key asset for the data driven approach in the BioFINDER-2 study, the extensive CSF measurements of 2944 proteins also limited the validation possibilities of the findings. Other cohorts with such extensive CSF measurements are difficult to reproduce and access due to financial and technical constraints. Additionally, the biological and technical variability of the suggested reference protein candidates should be further examined to ensure robustness of longitudinal measurements in participants. Furthermore, t-SNE is a data-driven method that with high probability will generate a very different visual result for a new dataset, limiting reproducibility. However, the actual clustering of proteins was semi-supervised (based on reference protein characteristics) and generated generalizable reference proteins, which we were able to confirm using a left-out BF2 test set and the external, independent cohort BF1. We also want to highlight that statistical associations in cross-sectional data were the focus in this work.

Conclusions about causal relationships would require further investigation. Lastly, we also want to acknowledge that the age of the BF2 cohort is relatively young compared to typical dementia patients. Nevertheless, this work supports the proof-of-principle that adjustment for non-disease related inter-individual differences in CSF protein levels will likely be highly useful in future studies aiming to understand associations between different CSF proteins or using key CSF proteins as diagnostic or prognostic biomarkers.

In conclusion, we show that non-disease related inter-individual differences in CSF protein levels affects diagnostic and prognostic performance for several CSF biomarkers. These differences can also result in false conclusions regarding associations between different CSF proteins or their relations to genetic variations. The issue can be addressed by using certain CSF reference proteins to represent the non-disease related concentration of the protein studied. Aβ40 is one of several promising general reference proteins (not just for Aβ42) and may be a suitable reference option due to its frequent availability in AD cohorts. Accounting for a CSF reference protein in future studies may make it possible to detect more proteins that are changed in a brain disease, and/or help ensure that reported correlations between CSF proteins are not mainly due to individual variability in CSF protein levels. Our reference protein method improves the accuracy of CSF biomarkers, and reduces the risk for false positive findings, with broad implications for both research and clinical practice.

## Methods

### Participants

Two study cohorts, both approved by the ethics committee at Lund University, were included: the Swedish BioFINDER-2 (BF2) cohort (enrollment from 2017 and still enrolling, $n = 982$, NCT03174938) and the Swedish BioFINDER-1 (BF1) cohort (enrollment between 2010 and 2015, $n = 1571$, NCT01208675). All participants were recruited at Skåne University Hospital and the Hospital of Ängelholm, Sweden. BF2 and BF1 consisted of individuals with either normal cognition (NC), subjective cognitive decline (SCD), mild cognitive impairment (MCI), dementia or another neurodegenerative disease. Conversion to AD dementia was determined during follow-up based on the treating physician's assessments[50]. Participants labeled as "non-converted" remained NC, SCD or MCI stable for at least two years. Further details about BF2 and BF1 can be found in ref. 51 and ref. 52 respectively, or at www.biofinder.se.

Only BF2 participants with complete CSF measures of 2944 proteins were included for analyses. No other exclusion criteria were implemented, but further filtering was later performed depending on the variables included in the statistical model. In BF1, the participants with complete CSF measures of 369 proteins were included in a similar manner.

### Ethics

The study was approved by the Swedish Ethical Review Authority. All participants gave their informed consent to participate in the study and the data were collected according to the Declaration of Helsinki.

### CSF collection and analysis

CSF samples were collected close in time after baseline examination (first visit) and handled according to established preanalytical protocols[51,53]. CSF samples were analyzed with validated, highly sensitive and specific Proximity Extension Assay (PEA) developed by OLINK Proteomics (Uppsala, Sweden). For BF2, the full OLINK Explore 3072 library was used, resulting in eight Proseek Multiplex panels (Oncology I and II, Neurology I and II, Cardiometabolic I and II, Inflammation I and II) to measure the concentration of 2943 CSF proteins. Each panel contained 367–369 proteins. For BF1, four panels (Neurology-exploratory, Neurology-I, Inflammation-I and Cardiovascular-III) were

used to measure the concentration of 368 CSF proteins. Each panel contained 92 proteins. All 368 proteins from the BF1 panels were also included in the BF2 Explore 3072 panels. Protein concentrations were provided as $\log_2$ scale of Normalized Protein eXpression (NPX) values. High NPX values correspond to high protein concentrations. Note that NPX values correspond to relative (and not absolute) protein concentrations within a cohort. A total CSF protein level was considered a less informative measure than a mean standardized CSF protein level in the context of (1) showing how brain-derived proteins follow a similar pattern and (2) when searching for reference proteins related to biomarkers for brain disease. This as total CSF protein levels would mainly reflect the most prominent proteins (albumin and other blood derived proteins[11]), making it unclear how less prominent proteins (brain-derived ones) reflect inter-individual variability. In contrast, a mean standardized CSF protein level weighs all proteins equally, making it robust to differences in absolute concentrations between proteins. Details of OLINK quality control and protein detectability are included in Supplementary Methods.

CSF biomarkers from the NeuroToolKit assay panel (P-tau181, Aβ42, Aβ40, sTREM2, YKL-40, GFAP, neurogranin, T-tau, S100, alpha synuclein and NfL) were measured in both cohorts using Elecsys assays in accordance with the manufacturer's instructions (Roche Diagnostics International Ltd)[54]. CSF analyses were performed by technicians blinded to all clinical and imaging data. CSF amyloid positivity was defined based on CSF Aβ42/Aβ40 that was dichotomized using the previously established cutoff of < 0.08[38].

## PET imaging

In both BF1 and BF2, amyloid-PET imaging was performed using [18F] flutemetamol. Standardized uptake value ratio (SUVR) images were created for the 90–110 min post-injection interval with whole cerebellum as reference region. A global neocortical composite region (volume of interest) corresponding to a set of cortical regions was used to summarize [18F]flutemetamol data, as described inref. 46. SUVR values were transformed into centiloids. The composite was used as a dichotomous variable with centiloids >20 regarded as amyloid positivity[55]. In BF2, tau-PET was correspondingly performed using [18F]RO948. SUVR images were created for the 70–90 min post-injection interval using the inferior cerebellar cortex as reference region. A composite corresponding to Braak I-IV regions[56] was used to represent AD-related tau pathology in the brain. The composite was used as a dichotomous variable with SUVR > 1.36 regarded as tau positivity[57].

## Statistical analysis

All analyses were implemented using Python version 3.9 or R version 4.2. When searching for reference proteins, all exploratory evaluations were performed using 10-fold-cross-validation within the BF2 training dataset. When evaluating results on the BF2 test dataset, the models were first refit on the full training dataset and thereafter evaluated once on the test dataset. When evaluating results on the BF1 dataset or a new model on the BF2 training dataset, bootstrap-resampling with replacement ($n_{iter} = 2000$) was performed such that a resampled training set of same size as the full dataset was created. Thereafter, a validation dataset was created from the participants that were never selected into the training set, which consequently varied in size between runs. This methodology was used to gain higher diversity between runs so that uncertainty estimations within a model could be performed with high reliability.

All protein concentrations were z-scored to allow for comparisons between measures in different units. Standardization was performed within each cohort and always fitted to training data. For every participant $i$, a mean standardized CSF level $y_i$ was computed as the average z-score over all 2943 OLINK proteins + Aβ40 for BF2, and all 368 OLINK

proteins + Aβ40 for BF1, Eq. 1:

$$y_i = \frac{1}{n_{proteins}} \sum_{j=1}^{n_{proteins}} z_j, \tag{1}$$

where $z_j$ is the z-score of protein $j$. Consequently, $n_{proteins} = 2944$ in BF2 and $n_{proteins} = 369$ in BF1. We investigated if a mean standardized CSF protein level calculated from the subset of 369 proteins (as was done in BF1) represented a similar structure as one that was calculated from 2944 proteins (as was done in BF2). In BF2, we computed a mean standardized CSF protein level for each participant using both the full sample (2944) and subsample (369) of proteins. The correlation coefficient between these two mean standardized CSF protein levels was 0.964, suggesting that the subset resulted in a similar representation of mean standardized CSF protein level as using all 2944 proteins.

For data exploration and visualization purposes, t-distributed stochastic neighbor embedding (t-SNE)[36] was applied. t-SNE is formulated as a non-linear optimization problem, aimed to preserve relative similarity of pairwise points in a high dimensional space when projected to a lower one[36]. The t-SNE results were combined with a semi-supervised K-means clustering[37] algorithm ($K = 20$) to sufficiently create subsets of data. The supervised part was performed by adjusting K and the random initialization seed so that clear structural clusters were separated and areas with characteristics relevant to a reference protein were joined. This was not a unique nor mathematically optimized way of dividing the t-SNE space. It was solely used due to its efficiency in this application as it enabled a more robust examination of subsets of similarly expressed proteins. To provide evidence that the results were not heavily dependent on the selection of K or random initialization, a robustness analysis is provided, see Supplementary Methods: K-means Robustness Analysis and Supplementary Figs. 14–16. Additionally, a sensitivity analysis when only using proteins with missing frequency <75% can be seen in Supplementary Methods: LOD Sensitivity Analysis and Supplementary Figs. 17–20, resulting in similar findings as when using all 2944 proteins.

To analyze associations with a continuous dependent variable, linear regression models (Supplementary Methods: Statistical models) were applied. In addition, partial correlation coefficients (Pearson) were computed to study correlation matrices between continuous variables. To predict a dichotomous variable, logistic regression models were applied (Supplementary Methods: Statistical models). As dichotomous data for most models was unbalanced, receiver operating characteristic (ROC) curve and Area under the ROC curve (AUC) were used to evaluate performance. AUCs were compared with a one-tailed ROC test using bootstrapping ($n_{iter} = 2000$). To compare AT(N)-groups, one-way ANCOVA was applied. *P* values were adjusted for multiple comparisons by Benjamini–Hochberg method. All models were adjusted for age and sex.

## Main models

To search for appropriate reference proteins, performance was evaluated in three logistic regression models. In each model, a well-established AD CSF biomarker (P-tau181 or Aβ42) was used as main predictor of either PET images or conversion to AD dementia. The three models were:

1. P-tau181→TauPET. Predicting tau-PET positivity with CSF P-tau181 as main predictor.
2. Aβ42→AβPET. Predicting Aβ-PET positivity with CSF Aβ42 as main predictor.
3. P-tau181→ADDconv. Predicting conversion to AD dementia versus remained stable for at least 2 years with CSF P-tau181 as main predictor.

The first two models were used to search for suitable reference proteins, the third was solely used for validation. The change in overall model performance when adjusting for different individual references was the measure of interest. A flowchart describing the pre-processing steps after split into training/test data for all models can be seen in Supplementary Fig. 21.

### Cell type expression, pathway enrichment and tissue expression analyses

To investigate properties of a cluster of proteins, we performed pathway enrichment and cell type expression analyses. For pathway enrichment, we used the WEB-based Gene SeT AnaLysis Toolkit (WebGestalt)[58]. We performed a human over-representation analysis (ORA) on cellular components, defining the background set as the 2943 OLINK proteins. For cell type expression, we used Seurat version 4.3.0 to analyze the open-access Human MTG 10x SEA-AD Allen Brain data from 2022[59]. This dataset includes single-nucleus transcriptomes from 166,868 total nuclei derived from the middle temporal gyrus (MTG) from five post-mortem human brain specimens. We used the class and subclass annotation available from the Allen Institute and applied the function AverageExpression (after removing all "None" annotations). From the average expression we then calculated a percentage expression across all cell types. A bootstrap enrichment test ($n = 10,000$) was used to compare significant (Benjamini–Hochberg corrected $q$ values $< 0.05$) cell type expression differences between a subset of proteins and all other proteins.

Tissue expression analyses were performed using three tissue datasets from the Human Protein Atlas (Normal tissue data, RNA consensus tissue gene data and RNA single cell type tissue cluster data, all found at: https://www.proteinatlas.org/about/download). Here, we examined (1) the proportion that had Medium/High level of detection in cerebral cortex or (2) the normalized Transcripts Per Million [nTPM] in cerebral cortex/brain, comparing proteins in cluster 11 ($n = 219$) to the rest of the highly detectable proteins ($n = 1512$).

### Reporting summary

Further information on research design is available in the Nature Portfolio Reporting Summary linked to this article.

## Data availability

Pseudonymized BioFINDER-1 and BioFINDER-2 data can be shared to qualified academic researchers after request (PI:OH) for the purpose of replicating procedures and results presented in the study. Data transfer must be performed in agreement with EU legislation regarding general data protection regulation and decisions by the Ethical Review Board of Sweden and Region Skåne. Human MTG 10x SEA-AD Allen Brain data from 2022[59] are publicly available and can be downloaded from celltypes.brain-map.org/rnaseq. Tissue datasets from the Human Protein Atlas are also publicly available and can be downloaded from https://www.proteinatlas.org/about/download. Source data are provided with this paper.

## Code availability

Code for the analyses can be found in the following GIT repository: https://github.com/karlssonlinda/reference_protein_project. Python dependencies include NumPy[60], pandas[61], Matplotlib[62], Scikit-learn[63], Statsmodels[64] and Pingouin[65]. R dependencies include Tidyverse[66] and pROC[67]. Seurat version 4.3.0 and WEB-based Gene SeT AnaLysis Toolkit (WebGestalt) 2019 were also used.

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

## Acknowledgements

Work at the authors' research center was supported by the Swedish Research Council (2022-00775), ERA PerMed (ERAPERMED2021-184), the Knut and Alice Wallenberg foundation (2017-0383), the Strategic Research Area MultiPark (Multidisciplinary Research in Parkinson's disease) at Lund University, the Swedish Alzheimer Foundation (AF-980907), the Swedish Brain Foundation (FO2021-0293), The Parkinson foundation of Sweden (1412/22), the Cure Alzheimer's fund, the Konung Gustaf V:s och Drottning Victorias Frimurarestiftelse, the Skåne University Hospital Foundation (2020-O000028), Regionalt Forskningsstöd (2022-1259), Swedish federal government under the ALF agreement (2022-Projekt0080), EU Joint Programme Neurodegenerative Diseases (2019-03401) and WASP and DDLS Joint call for research projects (WASP/DDLS22-066). The precursor of $^{18}$F-flutemetamol was sponsored by GE Healthcare. The precursor of $^{18}$F-RO948 was provided by Roche. Author JWV was supported by the SciLifeLab & Wallenberg Data Driven Life Science Program (grant: KAW 2020.0239). The funding sources had no role in the design and conduct of the study; in the collection, analysis, interpretation of the data; or in the preparation, review, or approval of the manuscript.

## Author contributions

L.K., O.H., N.M.C., J.V., I.A. and K.Å, designed the study. L.K. performed the analyses and data interpretations under the supervision of O.H., N.M.C., J.V., I.A. and K.Å. The manuscript was drafted by L.K. All authors contributed to preparation and critical review of the manuscript. O.H., S.J., K.B., S.P., E.S. and N.M.C collected the clinical data and/or coordinated/performed biomarker quantifications.

## Funding

## Competing interests

OH has acquired research support (for the institution) from ADx, AVID Radiopharmaceuticals, Biogen, Eli Lilly, Eisai, Fujirebio, GE Healthcare, Pfizer, and Roche. In the past 2 years, he has received consultancy/speaker fees from AC Immune, Amylyx, Alzpath, BioArctic, Biogen, Cerveau, Eisai, Eli Lilly, Fujirebio, Genentech, Merck, Novartis, Novo Nordisk, Roche, Sanofi and Siemens. KB has served as a consultant and at advisory boards for Acumen, ALZPath, BioArctic, Biogen, Eisai, Lilly, Novartis, Ono Pharma, Prothena, Roche Diagnostics, and Siemens Healthineers; has served at data monitoring committees for Julius Clinical and Novartis; has given lectures, produced educational materials and participated in educational programs for AC Immune, Biogen, Celdara Medical, Eisai and Roche Diagnostics; and is a co-founder of Brain Biomarker Solutions in Gothenburg AB (BBS), which is a part of the GU Ventures Incubator Program, outside the work presented in this paper. SP has acquired research support (for the institution) from ki elements / ADDF. In the past 2 years, he has received consultancy/speaker fees from Bioartic, Biogen, Lilly, and Roche. The remaining authors declare no competing interests.
