## [Peer Review File · Nature Communications]

Cerebrospinal fluid reference proteins increase accuracy and interpretability of biomarkers for brain diseasesREVIEWER COMMENTS

Reviewer #1 (Remarks to the Author):

This manuscript focuses on identifying ways to account for intra-individual differences in CSF AD biomarkers due to physiological reasons such as rate of CSF production or clearance in order to obtain a more accurate diagnosis. This is a notable problem and the manuscript and research presented are a clear addition to the field. My comments, mostly asking for clarifications, are listed below.

1. The average age of the Biofinder cohorts is young compared to typical dementia patients (e.g., BF2 training a mean of 68 years). Older patients typically have more pathology and are more likely to have NPH. The authors comment briefly on this in the discussion, but additional attention and acknowledgment of the younger age of the cohort is warranted.
2. Please provide the number of individuals who are CN, SCD, MCI, or have dementia in the results section.
3. It would be very informative to conduct a sensitivity analysis to determine whether results remain when stratified by diagnosis.
4. Could the authors clarify what is meant by “total CSF protein levels” and “mean CSF protein levels”? Also, does this vary for BF1 and BF2 because different proteins were examined? If this is correct, it would be helpful to describe it more clearly.

Reviewer #2 (Remarks to the Author):

This manuscript studies if models to predict amyloid PET status, tau PET status or progression to AD dementia with CSF ab42, ab40 and ptau181 measures can be improved by adding more proteins measured in CSF with the Olink platform, or by taking the average protein CSF values. One cluster of proteins was found that seems to improve the accuracy for PET status, whereas most other proteins did not. When taking one protein from that cluster as a ‘reference’ protein, individuals are reclassified from CSF amyloid and PET tau status, improving overall concordance from 76% to 89%.

The findings, that a particular group of proteins has higher CSF levels with abnormal tau, or decreased with normal tau seems to be in line with previous studies (e.g., references 1–4). The finding that ptau levels in CSF are related to amyloid metabolism, is also in line with previous studies, and most convincingly demonstrated in monoclonal anti-body trials that when plaques are removed t-tau and p-tau levels in CSF normalize (but not tau PET). The conclusion that the altered levels in these proteins are the result from increased protein production or reflect differences in CSF dynamics such as CSF volume such that lower CSF volume would increase protein levels is, however, not supported by the data presented. No measures for CSF dynamics or total protein levels were performed, and so these conclusions cannot be supported by the data presented in this study.

There are major limitations concerning the design and subsequent interpretation of the findings, which are potentially misleading:

1. In the introduction lines 68-72 make two assumptions: First that inter-individual variability in CSF biomarkers for AD pathology is caused by differences in CSF production or clearance rates. Second that these differences in CSF production and clearance rates are reflected in inter-individual differences in mean protein levels.

The manuscript continues to put forward these assumptions, but the validity of the assumptions is not tested:

Testing assumption 1 would require at least measures for CSF production and/or clearance rates, which in this study have not been measured. Ventricle size on MRI is considered in the manuscript, but the relationship between ventricle size and CSF dynamics is not tested (see e.g., ref 5 for an overview of methods on CSF dynamics). Furthermore, literature suggests that amyloid metabolism does not necessarily reflect CSF dynamics. For example, in Down Syndrome ab42 and ab40 are increased due to the presence of an additional APP gene, while CSF dynamics are normal(ref 6).

Testing assumption 2 would require for example measuring total protein in the CSF, which is a simple lab test but not performed in the present study. Instead, an average value of the Z-score of the NPX measures of Olink is taken. The NPX measures of Olink reflects, however, relative protein level differences, which do not directly translate to absolute concentrations (two proteins can have the same NPX value, but very different concentrations). These NPX measures were then scaled across the group, such that the average per protein is 0 with a standard deviation of 1. Moreover, if generic measures such as overall protein production or total CSF volume explain protein levels, then proteomic levels of all proteins should show the same pattern. Instead, this effect is restricted to one cluster of proteins. These proteins were associated with neuronal plasticity related processes. It could therefore also be argued that proteins that are associated with ab40 cluster together because they are part of a common pathophysiology of AD. Many other proteins are not measured with this technology, and so it remains unclear whether and how the average Olink values would be related to total protein or average protein levels, or indeed if this would be the case for other techniques as well.

2. Line 158-163 reports correlations of average protein levels with age, sex, and ventricular volume, which are presented as 'mechanisms' for average protein levels. These are correlations, and causality cannot be inferred. Age, sex and ventricular volume on MRI are variables, and it is unclear as to how these reflect causal mechanisms.

3. Line 321-323 reports that the correlation between a GMNC genetic variant with CSF ptau levels disappears, when correcting for ventricular volume. A suggestion is made that ventricular volume causes the relationship between a SNP and a protein. Ventricular volume can be altered due to a number of causes. A similar attenuation of the relationship between GMNC and ventricular volume would occur when correcting for ptau levels. It is not possible to infer causality from these correlations, and in particular the suggesting that ventricular volume would lead to altered CSF dynamics and thus to ptau levels cannot be derived from these analyses. Another indication that this is not a generic effect rises from the observation made that this influences specific proteins only. It is unclear why dilution of protein concentration in CSF would affect only a subgroup of proteins, and not all.

4. The implication stated on line 337 that future studies need to account for a reference protein, because otherwise results will be driven by average protein levels is not supported by the data presented. First, ab40 and all other proteins provided as a potential reference performed better than the average protein level only, suggesting that these proteins contain specific information. Second, only a

subgroup of proteins was increased with tau pathology, which indicates that the average protein levels per se do not explain the results. Indeed, this subgroup contributed to the mean protein levels, but from this finding it cannot be concluded that the specific group of proteins then reflect a generic trivial in- or decrease of protein levels, since other proteins did not show such a pattern, and since generic mechanisms were not tested in this study. Third, tissue proteomic studies have reported similar proteins as previous CSF proteomic studies to be associated with amyloid and tau pathology (e.g., refs 7,8). It is unclear how CSF dynamics, or average protein levels in the tissue would explain those results.

5. The implication posed on discussion line 384-387 is not supported by the data. It is not tested how ab40, as a marker for amyloid metabolism, can explain inter-individual differences in any other protein CSF biomarker considered. Furthermore, there is no biological rationale to support this statement. Whether or not a particular protein measured in CSF should be corrected by a reference protein should depend on 1. Which relationship is studied; 2. The mechanism that needs to be taken into account for a particular outcome. There is no evidence provided that amyloid metabolism is related to increased levels of all other proteins.

Other issues:

- Please add clinical status to the first 3 tables of table 1, and amyloid status to the bottom table.
- Supplementary table 2 should add another column to show if the Olink proteins improve on the model that also include ab40 levels. If prediction of PET status can be improved by ab40 only, this would be easier and cheaper than measuring Olink panels.
- Correlations as low as 0.24 are indicated as 'strong'. Please reserve the label strong for correlations >0.8.
- all figures and tables should include if there were any missing values in the Olink proteins reported (and if fully observed, a note in the legend would suffice).

1. Bader, J. M. et al. Proteome profiling in cerebrospinal fluid reveals novel biomarkers of Alzheimer's disease. *Molecular Systems Biology* 16, CD010783-17 (2020).
2. Sung, Y. J. et al. Proteomics of brain, CSF, and plasma identifies molecular signatures for distinguishing sporadic and genetic Alzheimer's disease. *Sci. Transl. Med.* 15, (2023).
3. Campo, M. del et al. CSF proteome profiling across the Alzheimer's disease spectrum reflects the multifactorial nature of the disease and identifies specific biomarker panels. *Nat Aging* 1–14 (2022) doi:10.1038/s43587-022-00300-1.
4. Visser, P. J. et al. Cerebrospinal fluid tau levels are associated with abnormal neuronal plasticity markers in Alzheimer's disease. *Mol Neurodegener* 17, (2022).
5. Liu, G., Ladrón-de-Guevara, A., Izhiman, Y., Nedergaard, M. & Du, T. Measurements of cerebrospinal fluid production: a review of the limitations and advantages of current methodologies. *Fluids Barriers CNS* 19, 101 (2022).
6. Atack, J. R., Rapoport, S. I. & Schapiro, M. B. Cerebrospinal fluid production is normal in Down syndrome. *Neurobiol. Aging* 19, 307–309 (1998).
7. Johnson, E. C. B. et al. Large-scale deep multi-layer analysis of Alzheimer's disease brain reveals strong proteomic disease-related changes not observed at the RNA level. *Nat Neurosci* 1–13 (2022) doi:10.1038/s41593-021-00999-y.
8. Higginbotham, L. et al. Integrated proteomics reveals brain-based cerebrospinal fluid biomarkers in asymptomatic and symptomatic Alzheimer's disease. *Sci Adv* 6, eaaz9360 (2020).

Reviewer #3 (Remarks to the Author):

Karlsson et al use the Biofinder 2 cohort to determine proteins, or groups of proteins, that can be used as reference proteins to adjust for, and improve interpretability of existing established CSF protein biomarkers, using a data driven approach. They are able to validate findings in a completely independent cohort (Biofinder 1). To search for relevant biomarkers they use regression models based on best gold standards available: tau pet, amyloid pet and clinical conversion to AD. They find that many established AD relevant protein biomarkers are influenced by mean CSF protein level, which may be important to adjust for in clinical and research biomarker interpretation. They also identify a number of individual protein biomarker candidates that individually serve as correction factors, and find some intriguing associations with ventricular volume.

Major comments

This is a highly novel and important study that shines a light on the important of adjusting for individual variations in CSF composition. Particularly as we enter an era of personalised medicine where understanding/ measuring/ targeting individual biological pathways will become standard practice.

The statistical methods are complex and would benefit from a formal statistical review, but the methods for identifying and validating biomarkers are clearly conveyed and seem sound.

The manuscript is extremely well written. Both the introduction and discussion are balanced and relevant.

Response Letter

We thank all reviewers for the time and effort that they have dedicated to this work and are grateful for the thorough feedback and insightful comments. We have considered all of the suggestions provided and incorporated what we think are suitable changes into our new version of the manuscript. Please find the changes in the manuscript/supplementary files and our detailed point-to-point response below (all changes tracked, and responses highlighted in orange font).

REVIEWER COMMENTS

Reviewer #1 (Remarks to the Author):

This manuscript focuses on identifying ways to account for intra-individual differences in CSF AD biomarkers due to physiological reasons such as rate of CSF production or clearance in order to obtain a more accurate diagnosis. This is a notable problem and the manuscript and research presented are a clear addition to the field. My comments, mostly asking for clarifications, are listed below.

1. The average age of the Biofinder cohorts is young compared to typical dementia patients (e.g., BF2 training a mean of 68 years). Older patients typically have more pathology and are more likely to have NPH. The authors comment briefly on this in the discussion, but additional attention and acknowledgment of the younger age of the cohort is warranted.

We agree with the reviewer that cohort properties are indeed always good to acknowledge. We have now made an additional comment regarding this in the discussion (lines 537-539: “*We also want to acknowledge that the age of the BF2 cohort is relatively young compared to typical dementia patients.*”). However, we believe that the results in this manuscript will generalize well, since inter-individual variability in CSF protein levels can be seen both for older and younger study participants (see figure below, comparing those above [right] and below [left] 70 years of age). We have added this figure in the Supplement with a reference to it in the main text (lines 180-181: “*The individual variability remained when stratifying on age and cognitive status [Supplementary Fig. 2 and 3].*”). Additionally, in all models in this work, we adjust for age and sex to ensure these variables are not driving results.

Supplementary Figure 2: Visualization of individual CSF levels stratified by age. As in Figure 2, for each participant (row) in the BF2 training set, the standardized concentration of 2,944 CSF proteins, sorted by increasing absolute association with the mean standardized CSF protein level is displayed. Systematic blue horizontal lines can be seen for individuals with consistently low values across most proteins, and correspondingly red horizontal lines for individuals with high values across most proteins (all relative to the total sample). a) below and b) above 70 years of age.

2. Please provide the number of individuals who are CN, SCD, MCI, or have dementia in the results section.

We thank the reviewer for pointing this out as we agree this makes the cohort description more informative for the reader. We have now added this information, see lines 155-159:

“The included participants had either normal cognition (NC, n=263 in BF2 and n=464 in BF1), subjective cognitive decline (SCD, n=111 in BF2 and n=172 in BF1), mild cognitive impairment (MCI, n=193 in BF2 and n=195 in BF1), dementia (n=216 in BF2 and n=73 in BF1) or another neurodegenerative disease (n=47, only in BF2).”

3. It would be very informative to conduct a sensitivity analysis to determine whether results remain when stratified by diagnosis.

We thank the reviewer for giving us an opportunity to expand on this as it is valuable to further confirm that results are not mainly driven by a specific diagnostic group. Therefore, we performed additional stratification analyses for several results in the manuscript.

Firstly, for the visualization of individual CSF levels (Fig. 2), we saw that the inter-variability remained within all stratified groups. This result further confirms that individual CSF variability was present in all cognitive groups and was non-disease

related. We have added this figure in the Supplement with a reference to it in the main text (lines 180-181: “*The individual variability remained when stratifying on age and cognitive status [Supplementary Fig. 2 and 3].*”):

Supplementary Figure 3: Visualization of individual CSF levels stratified by cognitive status.

As in Figure 2, for each participant (row) in the BF2 training set, the standardized concentration of 2,944 CSF proteins, sorted by increasing absolute association with the mean standardized CSF protein level is displayed. Systematic blue horizontal lines can be seen for individuals with consistently low values across most proteins, and correspondingly red horizontal lines for individuals with high values across most proteins (all relative to the total sample). a) NC b) SCD c) MCI d) Dementia e) Other neurodegenerative disease.

Secondly, we stratified the AUC scores for all predictive models (Fig. 5) in both BF1 and BF2. To maintain power for BF2, we bootstrapped the training set instead of evaluating the test set. Note that some of the stratified groups still were notably small, for example the dementia group (n=8) in the $A\beta_{42} \rightarrow A\beta_{PET}$ model. Additionally, as expected, some groups became very unbalanced. For example, only 13/234 NC were tau PET positive in the $P\text{-tau}_{181} \rightarrow \text{Tau}_{PET}$ model, and only 1 of 177 NC converted to AD dementia in the $P\text{-tau}_{181} \rightarrow \text{ADD}_{conv}$ model. These factors should be taken into consideration when reading the results in the figures below. Details of clinical status for all models (and hence size and balance for all stratified groups) have been added to Tab. 1 (as was suggested by reviewer #2).

The figures below suggest that a reference (not black) improves biomarker performance compared to no reference (black) also when stratified by cognitive status (and taking into consideration imbalance/low prevalence of subgroups). This is valid for both BF2 and BF1. Note that the range of the axes vary between subfigures for visualization purposes.

BF2

BF1

4. Could the authors clarify what is meant by “total CSF protein levels” and “mean CSF protein levels”?

We thank the reviewer for letting us clarify that these terms should not be used interchangeably. As we never use or refer to “total CSF protein levels” in this manuscript, we interpret that the reviewer is asking for a clarification of the term “mean CSF protein levels” rather than any specific adjustments. We will try to clarify this both in this response and in the manuscript.

The “mean CSF protein level” used here was computed from the OLINK + A β 40 measures using standardized protein concentrations (they all had mean value 0 and standard deviation 1). This makes all proteins “equally important” as they are not influenced by the proportion of the total CSF protein level that they make up (beneficial when searching for a trend common between proteins). The mean CSF protein level is also intuitive to interpret, as it represents how many standard deviations above or below the protein mean value a participant is on average, independently of what unit the protein concentration was measured in. That said, a more accurate term is “mean standardized CSF protein level,” which now is used throughout the manuscript. When appropriate, “mean CSF protein levels” has been replaced with “non-disease related CSF variability” or similar. Additionally, a clarification of these points has been added in the manuscript on lines 596-604:

“Note that NPX values correspond to relative (and not absolute) protein concentrations within a cohort. A total CSF protein level was considered a less informative measure than a mean standardized CSF protein level in the context of 1) showing how brain-derived proteins follow a similar pattern and 2) when searching for reference proteins related to biomarkers for brain disease. This as total CSF protein levels would mainly reflect the most prominent proteins (albumin and other blood derived proteins¹¹), making it unclear how less prominent proteins (brain-derived ones) reflect inter-individual variability. In contrast, a mean standardized CSF protein level weighs all proteins equally, making it robust to differences in absolute concentrations between proteins.”

Also, does this vary for BF1 and BF2 because different proteins were examined? If this is correct, it would be helpful to describe it more clearly.

In BF2, the mean standardized CSF protein level was created from the 2944 CSF proteins (2943 OLINK + A β 40). In BF1, the mean standardized CSF protein level was created from the 369 CSF proteins (368 OLINK + A β 40). Therefore, these average levels were indeed created from different numbers of proteins when comparing the two cohorts and we thank the reviewer for acknowledging that describing this more clearly improves the manuscript. We included as many proteins as we had available in each cohort to create this measure, because the more proteins, the more representative of the CSF inter-individual variability it should be. To confirm that these data differences between the two

cohorts did not have a notable impact on the computed mean standardized CSF level, we performed a sensitivity analysis using only the 369 proteins in BF1 to create a second mean standardized CSF protein level measure in BF2. We then compared it with the mean standardized CSF protein level measure from using all 2944 proteins. The correlation coefficient between a mean standardized CSF level of all 2944 proteins and the subset of 369 proteins (measured in BF1) was 0.964 in BF2. This suggests that the subset of 369 proteins measured in BF1 resulted in a similar representation of mean standardized CSF protein levels as the 2944 proteins measured in BF2. This is now also described in the methods section (lines 640-643), where we now write:

“The correlation coefficient between a mean standardized CSF level created from 1) all 2944 proteins and 2) a subset of the 369 proteins that were present in BF1, was 0.964 in BF2. This suggests that the subset of 369 proteins measured in BF1 resulted in a similar representation of mean standardized CSF protein level as using 2944 proteins in BF2.”

Reviewer #2 (Remarks to the Author):

This manuscript studies if models to predict amyloid PET status, tau PET status or progression to AD dementia with CSF ab42, ab40 and ptau181 measures can be improved by adding more proteins measured in CSF with the Olink platform, or by taking the average protein CSF values. One cluster of proteins was found that seems to improve the accuracy for PET status, whereas most other proteins did not. When taking one protein from that cluster as a ‘reference’ protein, individuals are reclassified from CSF amyloid and PET tau status, improving overall concordance from 76% to 89%

The findings, that a particular group of proteins has higher CSF levels with abnormal tau, or decreased with normal tau seems to be in line with previous studies (e.g., references 1–4). The finding that ptau levels in CSF are related to amyloid metabolism, is also in line with previous studies, and most convincingly demonstrated in monoclonal anti-body trials that when plaques are removed t-tau and p-tau levels in CSF normalize (but not tau PET). The conclusion that the altered levels in these proteins are the result from increased protein production or reflect differences in CSF dynamics such as CSF volume such that lower CSF volume would increase protein levels is, however, not supported by the data presented. No measures for CSF dynamics or total protein levels were performed, and so these conclusions cannot be supported by the data presented in this study.

We thank the reviewer for the thorough feedback and comments regarding this manuscript. Many of the points made by the reviewer we agree with and/or find important to investigate, and therefore we gratefully acknowledge that this revision has improved our manuscript.

However, we are afraid there might be some misunderstandings, and we might not have been clear in the manuscript. We would therefore like to highlight that in this manuscript, we consider the CSF proteins A β 42 and P-tau181 to be altered due to AD pathological changes (and not due to changes in e.g. CSF dynamics), which is why they are used as main predictors in several models. “Reference proteins”/”mean standardized CSF protein levels” are solely provided to improve these AD pathology-associated biomarkers, because non-disease related CSF inter-variability is considered an important confounder that makes the true associations to AD pathology less pronounced.

We agree that in this work we cannot prove that the cause of the non-disease-related individual differences is “CSF dynamics”. However, the hypothetical mechanism explaining the phenomenon we observe is not very relevant to the main conclusions of the manuscript. To further clarify, we would like to highlight that the main contributions of this work are the following (not related to any claims about CSF dynamics):

- 1) Non-disease related inter-variability in CSF protein concentrations exists (Fig 2).
- 2) Certain proteins can be used as references to represent non-disease related variations of brain-derived biomarkers in CSF (Fig 3-5).
- 3) This variability is a confounder that should be accounted for to improve the sensitivity and specificity of disease-related CSF biomarkers and avoid false positive findings of correlated CSF biomarkers (Fig 5-7). The latter is important as several previous articles (even in high impact journals) have reported findings that are likely confounded by non-disease related variations of brain-derived biomarkers in CSF (such as a large group of A-T+ individuals when using CSF A β 42 and CSF P-tau to define the AT groups, or associations between the levels of different CSF proteins).

Irrespective of what drives this phenomenon, our analyses clearly show that adjusting for a reference protein can improve AD CSF biomarker models. We have clarified our wording in the discussion’s first paragraph (lines 378-403) to match these three main contributions. It now reads:

*“We establish the existence of **non-disease related individual CSF variability (Fig. 2)**, which explain a considerable part of variation in CSF biomarkers. We conclude that many proteins are affected by individual CSF levels, including proteins relevant in AD, and may benefit from **adjustment of this confounding factor** when used as biomarkers. We identify a robust subset of potential reference proteins (“cluster 11”) from which we further characterize six specific reference protein candidates (*NTRK3*, *NTRK2*, *BLMH*, *CBLN4*, *PTPRN2* and *PTPRS*) that can significantly improve the accuracy of key AD biomarkers (**Fig. 3-4**). The results are validated on unseen test data and in an independent cohort (**Fig. 5**). We provide evidence that A β 40 works well as a reference protein, not only for A β 42, but for P-tau181 and other CSF biomarkers as well. Further, we show that several previously reported CSF biomarker classifications and associations **increased in accuracy/effect or** were greatly diminished when adjusting for reference proteins (**Fig. 5-7**). This implies that future studies should account for a reference*

protein to improve biomarker sensitivity and/or ensure that observed CSF biomarker relationships are not mainly driven by non-disease related differences in CSF protein levels. Our work focuses on biomarkers in AD, but since the issue of CSF variability is not AD specific, this concept likely has broad relevance across all neurological and psychiatric conditions where CSF biomarkers are used.”

There are major limitations concerning the design and subsequent interpretation of the findings, which are potentially misleading:

1. In the introduction lines 68-72 make two assumptions: First that inter-individual variability in CSF biomarkers for AD pathology is caused by differences in CSF production or clearance rates. Second that these differences in CSF production and clearance rates are reflected in inter-individual differences in mean protein levels. The manuscript continues to put forward these assumptions, but the validity of the assumptions is not tested:

Testing assumption 1 would require at least measures for CSF production and/or clearance rates, which in this study have not been measured. Ventricle size on MRI is considered in the manuscript, but the relationship between ventricle size and CSF dynamics is not tested (see e.g., ref 5 for an overview of methods on CSF dynamics). Furthermore, literature suggests that amyloid metabolism does not necessarily reflect CSF dynamics. For example, in Down Syndrome ab42 and ab40 are increased due to the presence of an additional APP gene, while CSF dynamics are normal(ref 6).

Testing assumption 2 would require for example measuring total protein in the CSF, which is a simple lab test but not performed in the present study. Instead, an average value of the Z-score of the NPX measures of Olink is taken. The NPX measures of Olink reflects, however, relative protein level differences, which do not directly translate to absolute concentrations (two proteins can have the same NPX value, but very different concentrations). These NPX measures were then scaled across the group, such that the average per protein is 0 with a standard deviation of 1.

We are appreciative of the reviewer’s comprehensive comment. First, we want to point out that there might be a misunderstanding about what is referred to as “assumption 1” by the reviewer. As we mention above, we do not believe that changes in CSF AD biomarkers are caused by differences in CSF dynamics. Actually, we postulated the opposite, that they are changed due to AD-related neuropathological changes in the brain parenchyma, but that the levels of these key proteins (e.g. A β 42 and P-tau) in CSF are to some degree confounded by inter-individual variations of other non-disease related phenomena. If such confounding exists, the association between CSF AD biomarkers and AD-neuropathology could actually be strengthened if models were appropriately adjusted for the confounding factor. Our first assumption is therefore that the diagnostic accuracy of disease-related markers (e.g. A β 42 and P-tau) is improved when their levels in CSF are adjusted by the CSF levels of reference proteins that actually do not change in neurological diseases (e.g. AD), but co-vary with the diagnostic biomarkers in healthy

individuals (where we could speculate that such associations are due to inter-individual variations in e.g., CSF dynamics).

When it comes to “assumption 2”, we interpret the reviewer’s comment that we should be more careful in concluding that the non-disease related variation in CSF concentrations of brain-derived proteins is due to inter-individual physiological variations in CSF dynamics. We thank the reviewer for challenging this hypothesis and acknowledging that certain formulations in the manuscript should be adjusted. We agree that from this work, we cannot prove, but can only speculate, that differences in “CSF dynamics” are behind the observed non-disease related inter-variability in CSF protein levels (which is observed in all diagnostic groups including healthy controls, as shown in the new Supplementary Fig 3).

Supplementary Figure 3: Visualization of individual CSF levels stratified by cognitive status.

As in Figure 2, for each participant (row) in the BF2 training set, the standardized concentration of 2,944 CSF proteins, sorted by increasing absolute association with the mean standardized CSF protein level is displayed. Systematic blue horizontal lines can be seen for individuals with consistently low values across most proteins, and correspondingly red horizontal lines for individuals with high values across most proteins (all relative to the total sample). a) NC b) SCD c) MCI d) Dementia e) Other neurodegenerative disease.

Therefore, we have adjusted some of the early phrasing so it is clear that inter-individual changes in CSF production and clearance is only one out of several possible explanations for this non-disease-related variation in proteins in CSF.

Lines 36-38 in the abstract:

“However, CSF biomarker concentrations may be influenced by non-disease related mechanisms which vary between individuals, such as CSF production and clearance rates.”

have now been changed to:

“However, CSF biomarker concentrations may be influenced by non-disease related inter-individual variability.”

Lines 71-83 in the introduction has been updated to:

*“However, the use of CSF biomarkers may be complicated by inter-individual variability in certain **non-disease related** physiological phenomena, **such as (but not limited to) subject-level variations in the transport rates of proteins from the brain parenchyma into the CSF or variations in CSF production/clearance rates.**⁸⁻¹⁰ Such inter-individual differences could lead to **non-disease related** differences in CSF protein levels¹¹, which could impact the overall performance of CSF biomarkers.¹²⁻¹⁴ Hypothetically, adjustment for individual **variability in CSF protein levels or to selected reference proteins** could optimize the performance of already efficient CSF biomarkers, reduce false positive findings (by attenuating biomarker associations that are driven by **non-disease related variability**), and increase the likelihood of making new biologically and clinically relevant discoveries. **Similar approaches are commonly applied in other areas of medicine. Biomarkers in urine are for example normalized to a reference marker (often creatinine) to adjust for variations in the urine concentration when the sample is collected at a single time point.**^{15,16}*

Line 123 in the introduction

“One striking example that highlights the potential of adjusting for processes related to CSF dynamics [...]”

has now been changed to:

“One striking example that highlights the potential of using a normalization protein [...]”

In line 137 in the introduction, “CSF dynamic” has been replaced with “confounding variability”.

Further, in the discussion, we have included the reviewer’s reference 5 and clarified that future studies are needed (that use e.g., these advanced MRI methods) to determine whether inter-individual variations in CSF production/clearance rates do explain why the use of reference proteins improves the diagnostic performance of key biomarkers like P-tau and A β 42. We now write (lines 441-444):

“We acknowledge that the hypothesis of CSF dynamics being a driving factor of inter-individual CSF variability needs to be confirmed in future studies before concluded, by for example examining relationships between the mean standardized CSF protein levels and CSF clearance and production rates.^{45”}

However, we again want to point out that the hypothesis about “CSF dynamics” is not central to this manuscript and does not affect the main finding that use of reference proteins improves the performance of key disease-related CSF biomarkers.

In terms of “total CSF protein levels”, we do not think this is a suitable measure in the current context. For the “total CSF protein level”, the concentration of all proteins present in a CSF sample should be measured. We avoid this measure because the “total CSF protein level” mainly consists of blood-derived proteins (~80%; especially albumin¹). Therefore, the “total CSF protein level” will to a larger degree reflect the integrity of the “blood to CSF barrier” (which can be estimated by the “CSF albumin to plasma albumin quotient”) and is therefore not as reflective of the brain-derived proteins of main interest in neurological diseases. In fact, in clinical practice, total CSF protein is often used in the diagnostic work-up of the *peripheral* neurological disease Guillain–Barré syndrome, where it provides very similar information as the “CSF albumin to plasma albumin quotient”.

Consequently, if one were to use total protein levels instead, the measure would mainly reflect the most prominent proteins (albumin and other blood derived proteins), making it unclear how less prominent proteins (brain-derived ones) reflect this inter-individual variability. This would not be optimal in the context of showing how brain-derived proteins follow a similar pattern even in healthy individuals and when searching for reference proteins related to biomarkers for brain disease. A clarification of this has been added in the manuscript on lines 596-604:

“Note that NPX values correspond to relative (and not absolute) protein concentrations within a cohort. A total CSF protein level was considered a less informative measure than a mean standardized CSF protein level in the context of 1) showing how brain-derived proteins follow a similar pattern and 2) when searching for reference proteins related to biomarkers for brain disease. This as total CSF protein levels would mainly reflect the most prominent proteins (albumin and other blood derived proteins¹), making it unclear how less prominent proteins (brain-derived ones) reflect this inter-variability. In contrast, a mean standardized CSF protein level weighs all proteins equally, making it robust to differences in absolute concentrations between proteins.”

1. Tumani, H., Huss, A. & Bachhuber, F. The cerebrospinal fluid and barriers – anatomic and physiologic considerations. in *Handbook of Clinical Neurology* vol. 146 3–20 (Elsevier B.V., 2017).

Moreover, if generic measures such as overall protein production or total CSF volume explain protein levels, then proteomic levels of all proteins should show the same pattern. Instead, this effect is restricted to one cluster of proteins. These proteins were associated with neuronal plasticity related processes. It could therefore also be argued that proteins

that are associated with ab40 cluster together because they are part of a common pathophysiology of AD. Many other proteins are not measured with this technology, and so it remains unclear whether and how the average Olink values would be related to total protein or average protein levels, or indeed if this would be the case for other techniques as well.

We would like to highlight that the findings of non-disease related (physiological) inter-individual variability in CSF protein levels are not solely restricted to the A β 40 cluster (Cluster 11), but proteomic levels of majority of proteins show the same pattern (especially highly detectable ones, shown in Supplementary Fig. 4, and further also seen in Fig. 2 and Supplementary Fig. 2 and 3).

Supplementary Figure 4: Visualization of individual CSF levels only including proteins with missing frequency < 75%. For each participant (row), the z-score of 1730 highly detected CSF proteins, sorted by absolute association with mean CSF level, is displayed. As expected, the individual CSF level is prominently shown in highly expressed proteins.

We confirmed in a new analysis that cluster 11 did not consist of proteins highly associated with AD with an ANOVA test. Here, we compared groups AD and NC, revealing that the proportion of proteins in cluster 11 significantly associated with AD was not notably higher than the rest of the 1512 highly detectable proteins (9% vs 7%, adjusting for age and sex, and FDR corrected).

2. Line 158-163 reports correlations of average protein levels with age, sex, and ventricular volume, which are presented as 'mechanisms' for average protein levels. These are correlations, and causality cannot be inferred. Age, sex and ventricular volume on MRI are variables, and it is unclear as to how these reflect causal mechanisms.

We agree with the reviewer that the data itself cannot reveal the causal relationship between these variables. We have therefore changed the phrasing “underlying mechanisms” to “factors associated with” in the manuscript (lines 189 and 421).

3. Line 321-323 reports that the correlation between a GMNC genetic variant with CSF ptau levels disappears, when correcting for ventricular volume. A suggestion is made that ventricular volume causes the relationship between a SNP and a protein. Ventricular volume can be altered due to a number of causes. A similar attenuation of the relationship between GMNC and ventricular volume would occur when correcting for ptau levels. It is not possible to infer causality from these correlations, and in particular the suggesting that ventricular volume would lead to altered CSF dynamics and thus to ptau levels cannot be derived from these analyses.

As the mentioned analysis does not include calculating correlations between CSF P-tau and GMNC genetic variant, we want to clarify this potential misunderstanding. P-tau is solely mentioned to highlight yet another case of possible false positive findings due to inter-individual variability, this time between this genomic region and several other CSF proteins. In this pQTL-analysis, proteins VEGFA, sVCAM1, PLXNB1, PRTG and TFF3 are selected to further investigate how a reference protein can affect other biomarkers. In summary, all these proteins show association of $\beta = 0.64-0.85$ with the mean standardized CSF protein level and their association with the GMNC-OSTN were markedly attenuated when adjusting for a reference protein. The results support that these reference proteins can attenuate false positive findings not only for models including P-tau and A β 42, but for other CSF biomarkers as well. A full description of this analysis can be found in *Supplementary Results: CSF pQTL Analysis (Association of genetic variants and proteins)* and Supplementary Tab. 6-7.

To further confirm the validity of these conclusions (and that amyloid pathological changes are not the driving factor), we performed an additional sensitivity analysis only including cognitively unimpaired A β -negative participants (n=395), see Table below. The results are very similar to those in Supplementary Tab. 7 (including all participants), confirming that these trans-pQTL associations of the GMNC-OSTN were markedly attenuated when adjusting for a reference protein also for cognitively unimpaired A β -negative individuals only (i.e. in healthy people).

Same as Supplementary Tab. 7 but only including unimpaired A β -negative participants (n=395).

	Without reference	With reference A β 40		With reference NTRK3	
	SNP	SNP	A β 40	SNP	NTRK3

Outcome Protein: VEGFA SNP: rs57712768	β-coef	0.18	0.10	0.61	0.012	0.86
	P-value	0.0002	0.008	4e-39	0.6	5e-115
	R-squared	0.16	0.47		0.78	
Outcome Protein: sVCAM1 SNP: rs146550622	β-coef	0.14	0.089	0.43	0.068	0.47
	P-value	0.004	0.03	1e-18	0.1	7e-24
	R-squared	0.20	0.36		0.39	
Outcome Protein: PLXNB1 SNP: rs4687181	β-coef	0.17	0.069	0.73	0.0075	0.87
	P-value	0.001	0.06	2e-56	0.8	7e-101
	R-squared	0.084	0.53		0.72	
Outcome Protein: PRTG SNP: rs71635338	β-coef	0.2	0.12	0.50	-0.001	0.88
	P-value	2e-4	0.007	8e-24	0.96	1e-133
	R-squared	0.12	0.32		0.77	
Outcome Protein: TFF3 SNP: rs78054167	β-coef	0.17	0.13	0.29	0.043	0.61
	P-value	4e-4	0.005	6e-9	0.2	6e-43
	R-squared	0.21	0.28		0.52	

Another indication that this is not a generic effect rises from the observation made that this influences specific proteins only. It is unclear why dilution of protein concentration in CSF would affect only a subgroup of proteins, and not all.

To further validate that the findings generalize to more proteins than the selected ones above, we performed an additional analysis testing all available CSF proteins in BF1 (n=368 when excluding the reference protein NTRK3) against the most common CSF *trans*-pQTL variant **rs71635338**. As seen in the figure below, almost all proteins showed severely weakened relationships with this SNP when adjusting for a reference protein. This figure has been added to the Supplementary Figures and is described in Supplementary Results:

“An additional analysis was performed for the most common CSF trans-pQTL variant rs71635338 testing its association with all available CSF proteins in BF1 (n=368 when

excluding the reference protein NTRK3), see Supplementary Fig 13. The results confirmed that adjusting for NTRK3 as a reference protein influenced not only specific proteins, but almost all tested proteins' relationship to this SNP diminished.”

Supplementary Figure 13: Effect of most common CSF *trans*-pQTL variant rs71635338 in associations with CSF proteins without and with reference protein NTRK3. The SNP rs71635338 is evaluated by association to each CSF OLINK protein in BF1 (n=368) in a linear regression model, without and with adjustment for the reference protein NTRK3. The difference is shown by β -coefficient (left) and P-value (right) for every protein. In general, the associations were severely weakened when adjusting for a reference protein. All models included BF1 participants (n=1445) and were adjusted for age, sex, dementia diagnosis and ten genetic principal components.

4. The implication stated on line 337 that future studies need to account for a reference protein, because otherwise results will be driven by average protein levels is not supported by the data presented.

We disagree with the reviewer that the implication that future studies need to account for a reference protein is not supported by the data presented. This manuscript provides several examples of why it is important that future CSF-based studies account for a reference protein when appropriate (including reproductions of previous studies that would have benefitted from addressing this confound). This we show in sections **Reference proteins explain discordance between CSF and PET tau positivity**, lines 327-354 and **Reference proteins often attenuate CSF biomarker associations**, lines 356-375, and the supplementary material referred to in these sections. We would here like to highlight for example the decreased discordance between CSF and PET tau positivity, where we found that the highly discussed CSF A-T+ group was reduced from n=37 to n=10 when using a reference protein (n=7 if grouping with PET). To avoid false positive findings in this highly discussed and researched group of individuals, it is important that there is a consensus between grouping techniques and that increased tau-levels are disease-related and not simply due to non-disease related CSF variability. Adding to this, we also showed how several previously reported results, including microglia-related CSF proteins (e.g. sTREM2, sAXL and sTyro3) being associated with CSF P-tau181 and elevated in the CSF A-T+ group, turned out to be substantially weaker (and more in line with PET) when accounting for a reference protein.

We have now included a more detailed explanation of our proposed reference protein method (lines 404-412):

“In general, when using disease-related biomarkers, we observe an altered level of a protein in a group of patients and relate it to a disease. Previously, the definition of “an altered CSF level” has been that the level in CSF is different from what is normal on a group level (for example when using a universal cutoff/threshold). Since we see inter-individual differences of the mean standardized CSF protein levels not related to disease (Fig 2 and Supplementary Fig. 2), this means that false positives (individuals with high biomarker levels not due to disease, but just because many protein concentrations in the CSF are relatively high) and false negatives (individuals with low biomarker levels but have the disease, but just because many protein concentrations in the CSF are relatively low) can occur from this commonly used method.”

Similar problems can be seen in many areas of medicine and have been overcome using reference markers. For example, markers in urine, like albumin, are commonly normalized to a reference marker (e.g., creatinine) to adjust for variations in the urine concentration when the sample is collected at a single time point. Similarly, when we measure proteins (or RNAs) in tissue, like brain tissue, the values of the target marker are normalized to a reference (a single marker like GAPDH or a composite of markers).

First, ab40 and all other proteins provided as a potential reference performed better than the average protein level only, suggesting that these proteins contain specific information.

It is not surprising to us that the reference protein candidates performed better than the mean standardized CSF protein level:

1) As mentioned in response to point 1) above, a large part of the CSF proteome is derived from the blood and therefore reflected in the mean standardized CSF protein level. The CSF concentrations of blood-derived proteins, like albumin, are likely affected by the integrity of blood-CSF barrier and thereby vary in concentration in a different way than disease-related brain-derived proteins, like A β 42 and P-tau. In our new analyses (lines 240-242): “we observed that the proteins of cluster 11 had higher brain expression than the other highly expressed OLINK proteins (Supplementary Tab. 2).”, further supporting this notion. This new analysis we describe in methods lines 705-710:

“Tissue expression analyses were performed using three tissue datasets from the Human Protein Atlas (Normal tissue data, RNA consensus tissue gene data and RNA single cell type tissue cluster data, all found at: <https://www.proteinatlas.org/about/download>). Here, we examined 1) the proportion that had Medium/High level of detection in cerebral cortex or 2) the normalized Transcripts Per Million [nTPM] in cerebral cortex/brain, comparing proteins in cluster 11 (n=219) to the rest of the highly detectable proteins (n=1512).”

The results are shown in Supplementary Tab. 2 and discussed in lines 524-529:

“Additionally, as a large proportion of the total CSF proteome consists of blood-derived proteins (~80%; mainly albumin¹¹), many CSF proteins are likely affected by the integrity of the blood-CSF barrier. As the proteins of cluster 11 showed higher brain expression than the other highly expressed CSF proteins, they are not as impacted by blood-CSF barrier permeability and therefore likely more robust reference proteins for biomarkers in brain diseases.”

Supplementary Table 2: Cluster 11 consisted of proteins that are more highly expressed in the brain compared to the other highly expressed CSF proteins. Row 1: the proportion that had Medium/High level of detection in cerebral cortex. Rows 2 and 3: the normalized Transcripts Per Million [nTPM] in cerebral cortex/brain, comparing proteins in cluster 11 (n=219) to the rest of the highly detectable proteins (n=1512). n_{tot} is the number of proteins that existed in that dataset.

	Cluster 11 (n=219)	Other proteins (n=1512)	Mann-Whitney U test result
Normal tissue data Tissue: cerebral cortex Unit: [Medium/High]	n _{tot} : 180 n _{elevated} : 131 73%	n _{tot} : 1109 n _{elevated} : 532 48%	
RNA consensus tissue gene data Tissue: cerebral cortex Unit: [nTPM]	n _{tot} : 217 mean: 73.3 CI: (0.8,206)	n _{tot} : 1494 mean: 48.8 CI: (0,154)	P-value = 7e-14

RNA single cell type tissue cluster data Tissue: brain Unit: [nTPM]	n_{tot} : 216 mean: 93.5 CI: (0,442)	n_{tot} : 1494 mean: 36.6 CI: (0,112)	P-value = 1e-13
--	--	---	-----------------

2) Further, as the 2944 CSF proteins measured in this work are involved in many different biological processes and can differ in e.g., size, charge, and solubility, it is not surprising that just taking the mean value of all 2944 proteins (and thereby not capturing any individual protein characteristics) was better than no reference but never the optimal reference for any biomarker. Instead, a protein that co-varies to a higher extent with the disease-related biomarker during normal physiology (non-disease conditions) would likely be an even better reference. This is why we searched for CSF proteins that correlate with CSF P-tau and A β 42 in cognitively unimpaired A β -negative participants only (Figure 3f and 3g). In the manuscript, we also provided in-depth information on the suggested reference protein candidates (see “Supplementary_Reference_Candidates”). Here, we saw that the proposed reference protein candidates were clearly more correlated with the mean standardized CSF protein level than the biomarkers we used as main predictors. Additionally, they were not predictive of the model outcomes without the main predictor in the model (AUCs between 0.62-0.67), indicating that they did not contain independent information but improved the models due to this confounding relationship.

As a comparison, one can again consider the similar application of references for urine biomarkers, which we now discuss in lines 486-493:

“Additionally, simply adjusting for the mean standardized CSF level never resulted in best performance, which also is in line with reference markers in other areas of medicine. As a comparison, one can consider how biomarkers in urine are processed today, where the normalization method of urine is critical due to dilution differences. For urine biomarkers, total protein concentration can be used as a standardization technique, but it is considered less precise compared to using ratios to other excreted small molecules or proteins like creatinine.^{15,16} This concept is very similar to what we propose with CSF reference proteins in this work.”

3) Good reference protein candidates should be reliably detectable from a methodology point of view, and e.g. exhibit levels well above the lower level of detection (LOD) on the majority of individuals. As many of the 2944 CSF proteins had a high missing frequency (see *LOD Sensitivity Analysis* in Supplementary Methods), the mean standardized CSF protein level was affected by this measurement noise. In Supplementary_Reference_Candidates, we observed that the reference protein candidates were indeed highly detectable in majority of subjects. We now added this to the following text (lines 261-267):

*“See Supplementary Reference Candidates for a biological description of the three proteins and BF2 data information further confirming the proteins’ potential as suitable references (e.g., high association with mean standardized CSF level, small concentration differences between diagnostic groups, high association with the main predictors, low model performance when used without a main predictor and **high detectability in majority of subjects [missing frequency ≤ 0.0008]).”***

Second, only a subgroup of proteins was increased with tau pathology, which indicates that the average protein levels perse do not explain the results. Indeed, this subgroup contributed to the mean protein levels, but from this finding it cannot be concluded that the specific group of proteins then reflect a generic trivial in- or decrease of protein levels, since other proteins did not show such a pattern, and since generic mechanisms were not tested in this study.

We are unsure what the reviewer means here. We do not observe (or claim) that reference proteins or the mean standardized CSF protein level is associated with tau pathology. It is CSF P-tau that is associated with AD pathology, but the association between CSF P-tau and AD pathology is “less strong” because of confounding effects of non-AD related physiological effects on the CSF proteome.

Further, we would like to highlight that not only the proteins of the cluster 11 subgroup, but most proteins that were highly expressed in CSF, varied in concordance with the mean standardized CSF protein level (as seen for example Supplementary Fig 4, but also in Fig. 2 and Supplementary Fig. 2 and 3) and are therefore affected by non-AD related physiological effects on the CSF proteome. This is also evident when comparing Fig 3b and 3c, where cluster 11 was one of several clusters covering the area of high association with the mean standardized CSF protein level.

Supplementary Figure 4: Visualization of individual CSF levels only including proteins with missing frequency < 75%. For each participant (row), the z-score of 1730 highly detected CSF proteins, sorted by absolute association with mean CSF level, is displayed. As expected, the individual CSF level is prominently shown in highly expressed proteins.

FIGURE 3: Dimensionality reduction reveals a cluster of CSF proteins with desired reference protein characteristics. T-distributed stochastic neighbor embedding (t-SNE),

reducing the high dimensional space of 658 participants to a two-dimensional one. Each scatter point illustrates one of the 2,944 CSF proteins, with relative similarity of pairwise proteins aimed to be preserved. In c)-h), min-max scaling for six different criteria has been performed to visualize relative differences within the space, all plots ranging between 0-1 (dark blue to yellow). In c)-f) associations are analyzed as absolute β -coefficients, all adjusted for age and sex. **a)** The raw t-SNE map. **b)** Semi-supervised K-means (K=20) clustering of t-SNE map, aiming to separate the evident clusters from t-SNE dimensionality reduction and areas of overlap in c)-h). t-SNE map colored by **c)** absolute association with mean standardized CSF level; **d)** absolute association with ventricle volume; **e)** absolute association with biomarker CSF P-tau181 (cognitively unimpaired A β -negative participants only); **f)** absolute association with biomarker CSF A β 42 (cognitively unimpaired A β -negative participants only); **g)** model performance when used as reference protein in **P-tau181**→**TauPET**; **h)** model performance when used as reference protein in **A β 42**→**A β PET**.

Third, tissue proteomic studies have reported similar proteins as previous CSF proteomic studies to be associated with amyloid and tau pathology (e.g., refs 7,8). It is unclear how CSF dynamics, or average protein levels in the tissue would explain those results.

We want to clarify that we do not claim that most previous findings from CSF studies not using reference proteins are wrong. We now write in the discussion (lines 413-417):

“Proteins that are majorly changed in CSF in a brain-disease like AD can clearly be identified even without a reference protein (like using CSF A β 42 without Ab40 as a reference). But when normalizing to reference proteins, we can likely detect more proteins that are changed in a brain disease and further avoid some false positive findings (as illustrated by many examples in this manuscript).”

Further, the diagnostic precision of key biomarkers (like p-tau) increases when adjusting for reference proteins (as illustrated by many examples our manuscript). We now write in the conclusion (lines 550-553):

*“Accounting for a CSF reference protein in future studies may **make it possible to detect more proteins that are changed in a brain disease, and/or help ensure that reported correlations between CSF proteins are not mainly due to individual variability in CSF protein levels.**”*

5. The implication posed on discussion line 384-387 is not supported by the data. It is not tested how ab40, as a marker for amyloid metabolism, can explain inter-individual differences in any other protein CSF biomarker considered. Furthermore, there is no biological rationale to support this statement. Whether or not a particular protein measured in CSF should be corrected by a reference protein should depend on 1. Which relationship is studied; 2. The mechanism that needs to be taken into account for a

particular outcome. There is no evidence provided that amyloid metabolism is related to increased levels of all other proteins.

We disagree with the reviewer that this implication is not supported by the data in the manuscript. We would again like to highlight sections **Reference proteins explain discordance between CSF and PET tau positivity**, lines 327-354 and **Reference proteins often attenuate CSF biomarker associations**, lines 356-375, which provide evidence that A β 40 can explain inter-individual variability for CSF biomarkers beyond A β 42 and P-tau181.

Further, we want to highlight that CSF A β 40 is in this work not regarded to be used as a marker of abnormal APP metabolism, but as a brain-derived marker that is continuously produced in the brain and virtually unchanged in CSF by neurodegenerative diseases including AD. As such, CSF A β 40 levels (like the levels of other CSF proteins) is influenced by inter-individual differences in physiological parameters affecting the CSF proteome (potentially for example transport of brain proteins into CSF, CSF production rate, CSF clearance rate etc). Comparing this to our previous example with the albumin/creatinine ratio in urine, creatinine is similarly used for studying abnormal permeability for albumin in the glomerulus. In this example, the target protein is albumin (like CSF A β 42 or P-tau in AD), but the reference protein is creatinine (like CSF Ab40 in AD), and the ratio is meant to reflect abnormal changes in albumin, and not creatinine. This because only urine levels of albumin are changed in diseases affecting the permeability of the glomerulus (like only CSF A β 42 and P-tau and not CSF Ab40 is changed in AD), and creatinine is used to normalize for non-disease related physiological changes in urine dynamics like inter- and intra-individual differences in the rates of urine production and clearance (similar to CSF Ab40 in brain/CSF).

We naturally considered A β 40 a potential reference protein candidate as it has previously been shown to improve the diagnostic accuracy of both A β 42 and P-tau181 (despite typically not being associated with AD on its own), something we also confirmed in this work. Additionally, it was found to have a stronger association ($\beta=0.44$, $P<1e-37$) with the mean standardized CSF protein level than the main predictors (P-tau181: $\beta=0.34$, $P<1e-20$, A β 42: $\beta=0.24$, $P<1e-10$), and it was not predictive of the model outcome if used independently (AUCs=0.62-0.65). This further confirmed that A β 40 improved the models due to this confounding relationship. A β 40 was only one of several potential reference protein candidates in this work but has the advantage that it has already been/will be measured in many AD cohorts. Therefore, it is potentially more feasible to adjust for A β 40 as a reference protein in relevant future works compared to the other candidates, which is why we highlighted this finding.

We agree with the reviewer that one should always consider potential confounding factors in a project-specific manner depending on what relationships are being studied. But if findings occur that are potentially confounded by inter-individual CSF protein level

differences, we hope that this work can assist in clarifying this, both through raising awareness of the phenomena, and by proposing reference protein candidates that can be used as a solution.

Other issues:

-Please add clinical status to the first 3 tables of table 1, and amyloid status to the bottom table.

We thank the reviewer for pointing this out, it has now been added to Table 1.

-Supplementary table 2 should add another column to show if the olink proteins improve on the model that also include ab40 levels. If prediction of PET status can be improved by ab40 only, this would be easier and cheaper than measuring Olink panels.

This is a very good point! However, we want to stress that our goal of showing that other CSF proteins (mainly neuronally-derived) also work as reference proteins is both a conceptual one (that a reference protein does not have to represent the same metabolic pathway as the target protein) and a practical one (if many different proteins can be used, it is easier for the researcher to find one that suits their situation). Therefore, the main aim of this work was not to improve models based on A β 40 as a reference when using OLINK-derived references. However, we have provided the suggested analysis below. These results show that combining A β 40 and one of the other reference protein candidates did not yield any significant improvements in model performances.

We would also like to mention that it is not necessary to measure the full OLINK panels to utilize these reference proteins. As we in this work have suggested a few possible reference protein candidates, cheaper methods (e.g., ELISAs) could be used to measure the concentration of each of these proteins separately instead.

	AUC A β 40 only (95% CI)	AUC OLINK reference protein only (95% CI)	AUC with A β 40 + OLINK reference protein (adjusted P -value compared to A β 40 only model)
P-tau181→TauPET (BF2)			
NTRK3	0.933 (0.896, 0.970)	0.920 (0.879, 0.962)	0.939 (0.17)
NTRK2	0.933 (0.896, 0.970)	0.923 (0.882, 0.964)	0.941 (0.14)
BLMH	0.933 (0.896, 0.970)	0.908 (0.861, 0.954)	0.937 (0.25)
CBLN4	0.933 (0.896, 0.970)	0.944 (0.907, 0.981)	0.948 (0.15)
PTPRN2	0.933 (0.896, 0.970)	0.937 (0.901, 0.974)	0.945 (0.14)
PTPRS	0.933 (0.896, 0.970)	0.928 (0.889, 0.967)	0.942 (0.17)

	Aβ42→AβPET (BF2)		
NTRK3	0.992 (0.982, 1)	0.983 (0.966, 1)	0.992 (0.71)
NTRK2	0.992 (0.982, 1)	0.987 (0.972, 1)	0.991 (0.71)
BLMH	0.992 (0.982, 1)	0.986 (0.971, 1)	0.992 (0.71)
CBLN4	0.992 (0.982, 1)	0.983 (0.966,1)	0.992 (0.71)
PTPRN2	0.992 (0.982, 1)	0.982 (0.965,1)	0.992 (0.71)
PTPRS	0.992 (0.982, 1)	0.983 (0.967,1)	0.992 (0.71)
	P-tau181→ADDconv (BF2)		
NTRK3	0.898 (0.814, 0.965)	0.920 (0.856, 0.971)	0.917 (0.21)
NTRK2	0.898 (0.814, 0.965)	0.912 (0.842, 0.967)	0.909 (0.21)
BLMH	0.898 (0.814, 0.965)	0.911 (0.843, 0.963)	0.910 (0.21)
CBLN4	0.898 (0.814, 0.965)	0.925 (0.866, 0.971)	0.920 (0.21)
PTPRN2	0.898 (0.814, 0.965)	0.925 (0.866, 0.974)	0.920 (0.21)
PTPRS	0.898 (0.814, 0.965)	0.922 (0.859, 0.975)	0.918 (0.21)
	P-tau181→ADDconv (BF1)		
NTRK3	0.933 (0.900, 0.962)	0.935 (0.906, 0.961)	0.943 (0.23)
NTRK2	0.933 (0.900, 0.962)	0.932 (0.901, 0.960)	0.944 (0.20)
BLMH	0.933 (0.900, 0.962)	0.896 (0.854, 0.937)	0.932 (0.45)
	Aβ42→AβPET (BF1)		
NTRK3	0.970 (0.938, 0.996)	0.932 (0.885, 0.974)	0.969 (0.81)
NTRK2	0.970 (0.938, 0.996)	0.931 (0.884, 0.971)	0.969 (0.81)
BLMH	0.970 (0.938, 0.996)	0.920 (0.870, 0.965)	0.969 (0.81)

-Correlations as low as 0.24 are indicated as ‘strong’. Please reserve the label strong for correlations >0.8.

We agree with the reviewer. The wording “strong” in the result and discussion sections has now been removed when referring to correlations <0.8.

- all figures and tables should include if there were any missing values in the Olink proteins reported (and if fully observed, a note in the legend would suffice).

As an initial data inclusion criterion for both BF1 and BF2 was “all with complete OLINK CSF protein and CSF A β 40 concentration measures” (as stated in lines 153-155), no missing OLINK values were present in any figures or tables.

1. Bader, J. M. et al. Proteome profiling in cerebrospinal fluid reveals novel biomarkers of Alzheimer’s disease. *Molecular Systems Biology* 16, CD010783-17 (2020).
2. Sung, Y. J. et al. Proteomics of brain, CSF, and plasma identifies molecular signatures for distinguishing sporadic and genetic Alzheimer’s disease. *Sci. Transl. Med.* 15, (2023).
3. Campo, M. del et al. CSF proteome profiling across the Alzheimer’s disease spectrum reflects the multifactorial nature of the disease and identifies specific biomarker panels. *Nat Aging* 1–14 (2022) doi:10.1038/s43587-022-00300-1.
4. Visser, P. J. et al. Cerebrospinal fluid tau levels are associated with abnormal neuronal plasticity markers in Alzheimer’s disease. *Mol Neurodegener* 17, (2022).
5. Liu, G., Ladrón-de-Guevara, A., Izhiman, Y., Nedergaard, M. & Du, T. Measurements of cerebrospinal fluid production: a review of the limitations and advantages of current methodologies. *Fluids Barriers CNS* 19, 101 (2022).
6. Atack, J. R., Rapoport, S. I. & Schapiro, M. B. Cerebrospinal fluid production is normal in Down syndrome. *Neurobiol. Aging* 19, 307–309 (1998).
7. Johnson, E. C. B. et al. Large-scale deep multi-layer analysis of Alzheimer’s disease brain reveals strong proteomic disease-related changes not observed at the RNA level. *Nat Neurosci* 1–13 (2022) doi:10.1038/s41593-021-00999-y.
8. Higginbotham, L. et al. Integrated proteomics reveals brain-based cerebrospinal fluid biomarkers in asymptomatic and symptomatic Alzheimer’s disease. *Sci Adv* 6, eaaz9360 (2020).

Reviewer #3 (Remarks to the Author):

Karlsson et al use the Biofinder 2 cohort to determine proteins, or groups of proteins, that can be used as reference proteins to adjust for, and improve interpretability of existing established CSF protein biomarkers, using a data driven approach. They are able to validate findings in a completely independent cohort (Biofinder 1). To search for relevant biomarkers they use regression models based on best gold standards available: tau pet, amyloid pet and clinical conversion to AD. They find that many established AD relevant protein biomarkers are influenced by mean CSF protein level, which may be important to

adjust for in clinical and research biomarker interpretation. They also identify a number of individual protein biomarker candidates that individually serve as correction factors, and find some intriguing associations with ventricular volume.

Major comments

This is a highly novel and important study that shines a light on the importance of adjusting for individual variations in CSF composition. Particularly as we enter an era of personalised medicine where understanding/ measuring/ targeting individual biological pathways will become standard practice.

The statistical methods are complex and would benefit from a formal statistical review, but the methods for identifying and validating biomarkers are clearly conveyed and seem sound.

The manuscript is extremely well written. Both the introduction and discussion are balanced and relevant.

We thank the reviewer for an accurate summary of this manuscript and for emphasizing the importance and potential of accounting for CSF inter-individual variability.

Reviewers' comments:

Reviewer #1 (Remarks to the Author):

The authors have addressed all of my comments.

Reviewer #2 (Remarks to the Author):

In this revised version, some of the wording in the manuscript is altered. Like in the previous version, the only claims supported by the data is that individuals vary in their CSF protein levels, and that predicting amyloid or tau PET with CSF markers for amyloid and tau can improve when including an additional protein in the model. All claims and interpretations beyond what was tested cannot be supported by the present data. In particular (but not limited to what is listed) the conclusions that:

- associations of previous CSF proteins with other outcomes than PET, such as genetic variants or other proteins, were spurious because the authors show that these associations can change when including the reference protein that correlates with outcomes of interest. This is a classic example of collinearity. Without a clearly defined mechanism it cannot be determined from correlation analyses if 1 protein is the confounder, while the other protein associations are 'true'. Such collinearity can lead to spurious positive findings, switching of signs, or false negative findings. Without a mechanism as to how a particular protein is related to a particular outcome (e.g., ptau and GMNC) any reference to causality should be avoided. This also means, that although authors don't test ptau and GMNC (while data is available), the notion that associations of other proteins with GMNC change after adjusting analyses for a reference protein that also correlates with those proteins and/or the outcome, means that previous results were 'spurious' is too strong. When there is no clear mechanistic relationship between the reference and e.g., GMNC, it is impossible to conclude that the findings were spurious based on these correlation patterns only.

- Another example of collinearity is s-figure 13: it is unsurprising that the proteins lose their correlation with the outcome when NTRK3 is also included in the model if they correlate with NTRK3 (and or if NTRK3 also correlates with the outcome), while the non-correlated proteins now are correlated. This is probably also a collinearity effect.

- the new version now interprets inter-individual variability to be 'non-disease related'. This is another strong notion that is not tested, but rather based on the observation that some proteins show similar between subject variability in controls as in AD groups. This may mean that those proteins may indeed not be related to amyloid and tau, but the possibility cannot be excluded that such proteins may still play a role in the disease. For example, APOE plays a strong role in AD pathogenesis, and specific APOE fragments reflecting the different types of alleles are altered before amyloid is altered. According to the reasoning of the authors, this would mean that APOE is 'non-disease related'.

Reviewer #3 (Remarks to the Author):

The authors have satisfactorily dealt with the comments raised by the other referees.

We thank the reviewer for sharing concerns about the methodological approach and interpretations of our manuscript. To address these, we performed new analyses providing evidence that:

- 1) Parameter estimates from our models are trustworthy and not influenced unduly by collinearity.**
- 2) GMNC associations with P-tau is congruent with the other GMNC results of this manuscript.**
- 3) Results are quite comparable whether we use single reference proteins or the mean standardized CSF protein level (created from ~3000 proteins), suggesting reference proteins are representative of inter-individual variations in the concentrations of the CSF proteome.**

For further details on each of these points, please see our extended response below.

In this revised version, some of the wording in the manuscript is altered. Like in the previous version, the only claims supported by the data is that individuals vary in their CSF protein levels, and that predicting amyloid or tau PET with CSF markers for amyloid and tau can improve when including an additional protein in the model.

Authors' response: The reviewer seems to agree (“claims supported by the data”) with our important main finding that there is individual variability in CSF protein levels, and that CSF biomarker ability to predict amyloid or tau PET is improved when adjusting for additional proteins. These are our main findings in this manuscript, all supported by data and seen in Figures 2-6.

An important correction to the Reviewer’s comment – we find the ability of CSF biomarkers to predict PET pathology is not only improved when adding single additional proteins (reference proteins), but also when using a mean signal from ~3000 proteins in the CSF, which is a measure aimed to robustly reflect the individual CSF protein variability (we show this in main text **Fig. 5**, which we have included below for convenience). We showed that single reference proteins that improve prediction models were also highly correlated with this mean CSF signal, yet conveniently summarize this variability with a single marker (easier to reproduce).

FIGURE 5: Performance evaluation of all models with and without references. ROC curves in a) and b), and corresponding AUCs in c) and d), for the two models **P-tau181**→**TauPET** and **Aβ42**→**AβPET**, evaluated on the BF2 test dataset. AUCs for e) model **P-tau181**→**ADDconv** on the training dataset BF2, f) model **P-tau181**→**ADDconv** on BF1 and g) model **Aβ42**→**AβPET** on BF1. No reference corresponds to the model without use of a reference protein, consistently outperformed by all tested references: Aβ40, mean standardized CSF level, NTRK3, NTRK2, BLMH, SVD1, SVD2, CBLN4, PTPRN2, PTPRS and SVD2. Quantitative details can be found in Supplementary Tab. 3. For visualization purposes, the ROC curves do not show all protein candidates but solely the corresponding SVDs.

* $P < 0.05$, ** $P < 0.01$ compared to the bar of same color as asterisk.

All claims and interpretations beyond what was tested cannot be supported by the present data.

In particular (but not limited to what is listed) the conclusions that:

- associations of previous CSF proteins with other outcomes than PET, such as genetic variants or other proteins, were spurious because the authors show that these associations can change when including the reference protein that correlates with outcomes of interest. This is a classic example of collinearity. Without a clearly defined mechanism it cannot be determined from correlation analyses if 1 protein is the confounder, while the other protein associations are ‘true’. Such collinearity can lead to spurious positive findings, switching of signs, or false negative findings. Without a mechanism as to how a particular protein is related to a particular outcome (e.g., ptau and GMNC) any reference to causality should be avoided.

Authors’ response: We understand most of the reviewer’s feedback pertains to the GMNC analyses, which are ultimately a minor part of this manuscript. However, we must note that the results when using GMNC instead of PET applied the same method and, as we will show below, there is no evidence of collinearity present in our models.

A confounder is defined as a variable associated with both the predictor and outcome of interest, where not modeling it can lead to false positive or false negative findings. The fact that the main

predictor and reference protein are correlated is therefore expected but does not necessitate the specific case of collinearity. However, model collinearity can be tested objectively. We now include an analysis formally testing for collinearity in our models and we see no evidence that our models violate collinearity assumptions:

We tested all 368 multivariate regression models in Supplementary Fig. 13, using the Variance Inflation Factors (VIF) to test for multicollinearity between the main protein and reference protein (in this case, NTRK3). For VIF, values between 1 and 5 are considered moderate correlation that do not violate collinearity assumptions. The histogram below shows that, of the 368 models included, only 15 (4%) have VIF >5 for either the reference protein or main protein tested. These analyses suggested rather unambiguously that parameter estimates from our models are trustworthy and not influenced unduly by collinearity.

This also means, that although authors don't test ptau and GMNC (while data is available), the notion that associations of other proteins with GMNC change after adjusting analyses for a reference protein that also correlates with those proteins and/or the outcome, means that previous results were 'spurious' is too strong. When there is no clear mechanistic relationship between the reference and e.g., GMNC, it is impossible to conclude that the findings were spurious based on these correlation patterns only.

Authors' response: As the reviewer mentions, we did not previously include P-tau in these analyses. Instead, we included other proteins that have been shown by our group (ref. 1) to be associated with GMNC (and we clearly here acknowledge that our previous findings can be questioned in the light of our new results). However, since associations between a GMNC variant and CSF P-tau also has been suggested in previous works (e.g., ref. 2), we have now also included CSF P-tau in these analyses. The results are:

No reference:

CSF P-tau ~ GMNC + covariates → $\beta = 0.078$, p-value 0.0066*

With reference:

CSF P-tau ~ GMNC + mean standardized CSF protein level + covariates → $\beta = 0.017$, p-value 0.53

CSF P-tau ~ GMNC + CSF A β 40 + covariates → $\beta = 0.031$, p-value 0.23

CSF P-tau ~ GMNC + CSF NTRK3 + covariates → $\beta = 0.0053$, p-value 0.84

In conclusion, CSF P-tau was significantly associated with the tested GMNC variant rs71635338 when not accounting for inter-individual CSF variability, but is no longer significant when adjusting for multiple markers of inter-individual CSF variability described in our paper (including simply using the mean standardized protein level). These results are congruent with the other GMNC analyses in this manuscript.

Statistical analyses above and below suggest that the GMNC relationship with P-tau (and other proteins) might be explained by a separate mechanism. Since GMNC variants affect ventricular volume and show strong associations with many CSF proteins, we speculate that these relationships are due to dilution properties related to CSF volume and strengthen this with the fact that these relationships are severely reduced when adjusting for a reference protein (or the mean standardized CSF protein level, see below). The reviewer is correct that we cannot determine causality from these analyses, since they are associative (as were the original findings, ref 2). However, our analysis shows that the confounding influence of CSF variability on the relationship between GMNC and P-tau (as well as other CSF proteins) cannot be ruled out, and cannot be explained by collinearity, which we believe is a finding that should be reported.

We did not include the GMNC – P-tau analysis in the previous version of our manuscript. The reviewer's comment seems to suggest this as an omission, so we would be happy to include this in the updated version. Since the Reviewer seems to take issue with the GMNC analysis generally, we can also remove this finding entirely from the manuscript if the editor agrees with this decision. We do not believe this would be warranted by any methodological issue, but the finding is ultimately only a minor part of the paper that we do not consider worth prohibiting the outreach of our main results.

- Another example of collinearity is s-figure 13: it is unsurprising that the proteins lose their correlation with the outcome when NTRK3 is also included in the model if they correlate with NTRK3 (and or if NTRK3 also correlates with the outcome), while the non-correlated proteins now are correlated. This is probably also a collinearity effect.

Authors' response: We understand the reviewer's concerns. We would like to highlight the new VIF analysis above, which confirms that collinearity assumptions are not violated in these models. Additionally, we now report two new sensitivity analyses of Supplementary Fig. 13, yielding very similar results as the original approach:

- 1) using the mean standardized CSF protein level as reference (to address the concern that the reference protein may not represent the inter-individual CSF variability but may have its own relationship with the outcome)
- 2) using sequential regression (regressing out the reference protein from the main predictor before applying it in the multivariate linear regression model, another way to address potential collinearity)

First, here is the original figure for reference:

Supplementary Figure 13: Effect of most common CSF *trans*-pQTL variant rs71635338 in associations with CSF proteins without and with reference protein NTRK3. The SNP rs71635338 is evaluated by association to each CSF OLINK protein in BF1 (n=368) in a linear regression model, without and with adjustment for the reference protein NTRK3. The difference is shown by β -coefficient (left) and P-value (right) for every protein. In general, the associations were severely weakened when adjusting for a reference protein. All models included BF1 participants (n=1445) and were adjusted for age, sex, dementia diagnosis and ten genetic principal components.

Here, we show a similar effect when using mean standardized CSF protein level

Effect of most common CSF *trans*-pQTL variant rs71635338 in associations with CSF proteins without and with reference mean standardized CSF protein level. Otherwise same as Supplementary Fig. 13.

And here, we show a similar effect once again when using sequential regression:

Effect of most common CSF *trans*-pQTL variant rs71635338 in associations with CSF proteins without and with reference NTRK3 in a sequential regression (regressing out reference protein beforehand). Otherwise same as Supplementary Fig. 13.

- the new version now interprets inter-individual variability to be ‘non-disease related’. This is another strong notion that is not tested, but rather based on the observation that some proteins show similar between subject variability in controls as in AD groups. This may mean that those proteins may indeed not be related to amyloid and tau, but the possibility cannot be excluded that such proteins may still play a role in the disease. For example, APOE plays a strong role in AD pathogenesis, and specific APOE fragments reflecting the different types of alleles are altered before amyloid is altered. According to the reasoning of the authors, this would mean that APOE is ‘non-disease related’.

Authors’ response: We do claim that the inter-individual *variability* we observe is non-disease related, and we support this claim with data that the variability exists independent of diagnosis. We agree with the reviewer that there could still be disease-related differences in the mean levels of certain individual proteins, even if the inter-individual variability is similar. To further confirm that the mean standardized CSF protein level is not significantly different between diagnostic groups, we performed an ANCOVA analysis formally testing this (adjusted for age and sex, and FDR corrected):

The mean standardized CSF level did not significantly differ between diagnostic groups. *P*-values (adjusted for age and sex and FDR corrected) assessed by a one-way ANCOVA comparing differences in mean standardized CSF protein levels between diagnostic groups. This analysis included BF2 training set participants, split by diagnosis: normal cognition (NC), subjective cognitive decline (SCD), mild cognitive impairment (MCI), Alzheimer dementia (AD) or another neurodegenerative disease (OD).

	NC vs SCD	NC vs MCI	NC vs AD	NC vs OD	SCD vs MCI	SCD vs AD	SCD vs OD	MCI vs AD	MCI vs OD	AD vs OD
Mean standardized CSF protein level	0.12	0.90	0.72	0.93	0.20	0.12	0.12	0.73	0.93	0.66

In conclusion, no significant differences of mean standardized CSF protein levels were seen between diagnostic groups.

Regarding the APOE argument, we are not sure what the reviewer is suggesting. There are differences between controls and AD in "specific APOE fragments": AD have higher CSF ApoE4 levels than controls have, and they also have more APOE e4 alleles, so we would not reason that APOE is non-disease related.

1. Hansson, O. *et al.* The genetic regulation of protein expression in cerebrospinal fluid. *EMBO Mol Med* **15**, (2023).
2. Jansen, I. E. *et al.* Genome-wide meta-analysis for Alzheimer's disease cerebrospinal fluid biomarkers. *Acta Neuropathol* **144**, 821–842 (2022).

REVIEWER COMMENTS

Reviewer #2 (Remarks to the Author):

For point 1 and related responses: Thank you for calculating the VIF factors. It remains unclear from these analyses how the VIF values actually relate to these changes in beta's displayed in supplementary figure 13: Here some beta's decrease, whereas other beta's increase or remain stable. From the histogram it is not possible to determine whether this may or may not have had an effect. Thus, although it is reassuring that the authors show that VIF effects are not excessively high for most variables tested, it remains unclear if there is an association between particular beta's and higher VIF. The alternative approach that first regresses out the reference protein from the outcome, is indeed not identical to directly including both reference and SNP in the same model, but the idea remains the same, i.e., to test what part of the variance in the outcome is explained by the SNP after correcting within the model (assuming that type III variance testing was performed here), or before. Yes, it is clear that there is a specific group of proteins that seems to correlate with each other, with disease processes and with the genetic variant. The present analyses indicate that the correlations do not lead to excessive VIF values, but to state that thus the relationship between a genetic variant and protein levels as outcome are false cannot be concluded from the analyses (see next point).

For point 2 and related responses: From a conceptual point of view the interpretation of these analyses remains flawed, and is not resolved with the new analyses that the authors present: the reference proteins were selected to be correlated with p-tau levels, but their mechanistic relationship with the GMNC-OST SNP remains unclear and cannot be determined based on the present data. A confounding variable should influence both the outcome as well as the other predictor. However, in this case the other predictor is a genetic variant, it cannot be assumed that the reference protein levels would influence the genetic variant, as a true confounding variable should do. Other alternative explanations may be conceptually more likely: either this genetic variant influences the levels of both the outcome as well as the reference proteins directly or indirectly via another mechanism that is related to all proteins tested here, i.e., the relationship between reference and outcome protein are both downstream effects of the genetic variant, or alternatively, that the reference protein levels themselves are influenced by the mechanism related to tau tangling (and thus p-tau levels) as well as by the genetic variant, which would indicate collider bias. Although, there is speculation on potential mechanisms, such as the role of ventricular volume, those mechanisms are not tested in the present manuscript. Given that the VIF were indeed not excessively high for most proteins in supplementary figure 13, another possibility could also be that these proteins adjusted for the disease effect, since these genetic correlations were tested across the group with both controls and AD individuals, 'correcting' for a variable related to tau (and possibly amyloid pathology) would correct for the disease effect on correlated proteins. The authors present in their rebuttal an ANOVA analyses, comparing the mean Z-scores across proteins in individuals between diagnostic groups, and the MCI, AD and OD groups show on average much larger levels than controls (there were no p values to be found in my version of the rebuttal; but given that these are Z scores there is almost a standard deviation difference between controls and MCI/dementia), which is quite a lot. It is unclear why this comparison is made based on

clinical status, while table 1 in the main manuscript indicate that roughly half the SCD and MCI groups are amyloid positive and the other amyloid negative, similarly for tau status. The reference proteins and average Z scores were related to those pathologies so why lump negative and positive together? Thus, from those additional analyses it cannot be concluded that there is no disease effect on the average protein levels (which happen to correlate with tau and/or amyloid?). The models presented based this data cannot distinguish whether the reference proteins are indeed true confounders, colliders or are all downstream effects of the genetic variant. Which brings me back to the issue previously raised with these analyses, which are indeed not the main analyses in the present manuscript, but still these results are used by the authors to make generalising implications that per default all CSF proteomic analyses require correction with reference proteins. This suggestion remains unsupported by the present data. Whether or not to adjust analyses for a particular confounder should be determined on the hypothesis tested and not based on these analyses only. Therefore, the conclusion provided by the authors remains unsupported by the data and analyses performed, in particular the genetic part: “We show that non-disease related inter-individual differences in CSF protein levels confounds diagnostic and prognostic performance for several CSF biomarkers. These differences can also result in false conclusions regarding associations between different CSF proteins or their relations to genetic variations. The issue can be addressed by using certain CSF reference proteins to represent the non-disease related concentration of the protein studied.” What can be concluded based on the present analyses is that certain genetic variants may influence the levels of specific groups of proteins. Whether such associations are ‘false’ cannot be based on these analyses. Indeed, if these analyses were not part of the main objectives, and causality cannot still not be inferred with the present data, then why present these results in such a way?

For point 3 and related responses: The authors argue that reference proteins are representative of inter-individual variation in CSF protein concentrations, seems circular: that results remain comparable when just taking the Z score average of all protein levels instead of a single protein in the present data set, but the reference protein with about a third of proteins had a similar correlation pattern across individuals and thus influence the average Z-score (they are not independent measurements). Most relationships are attenuated with average Z-score, which may reflect that this indeed also includes proteins that do not correlate with e.g., tau, unlike the selected reference proteins. It remains unresolved what the underlying mechanism may be other than that a specific group of proteins changes similarly across the disease.

To further clarify the point on APOE, which was only a detail: APOE was taken as an example to point out that it is not a trivial task to determine whether protein levels may or may not be disease related, which in this manuscript was based on having normal cognition and no AD pathology. APOE genotype has effects on CSF protein levels even before people have dementia, and indeed even before people have abnormal amyloid. According to the rationale presented in this manuscript, where non-disease related is based on finding altered protein levels in controls as well, this would disqualify APOE as being related to the disease (which clearly is related to the disease, which the authors also acknowledge).

Reviewer #4 (Remarks to the Author):

[Editorial Note: See PDF on next page]

Reviewer's report for "Cerebrospinal fluid reference proteins increase accuracy and interpretability of biomarkers for brain diseases"

The authors address an important issue of non-disease related inter-individual variability in cerebrospinal fluid (CSF) protein levels and its potential impact on the diagnostic accuracy of Alzheimer's disease (AD) CSF biomarkers. The authors provide a data driven approach to identify potential reference proteins that can significantly improve the accuracy of AD biomarkers. I have a few questions or comments for further clarification on statistical application.

Comments:

- If I understood properly, mean standardized CSF level was computed as the average z-score over all 2,943 + A β 40 for BF2 and all 368 OLINK proteins + A β 40 for BF1. For researchers who are going to work on data like this, having a clear picture would be helpful. For instance, letting y_i as the dichotomous response and denoting $z_1, z_2, \dots, z_{np_{BF2}}$ as the z-scores corresponding to 2943 proteins + A β 40 for BF2, where number of proteins, $c = 2944$. Then, mean standardized CSF for the i th subject is calculated as $\frac{1}{np_{BF2}} \sum_{j=1}^{np_{BF2}} z_j$. Similar or convenient notation can be used for BF1 (e.g., $np_{BF1} = 369$).

Here $np_{BF2} \ll np_{BF1}$, in this case, could the author justify BF1 data represent similar structure as for BF2?

- "The correlation coefficient between a mean standardized CSF level created from 1) all 2944 proteins and 2) a subset of the 369 proteins that were present in BF1, was 0.964 in BF2. This suggests that the subset of 369 proteins measured in BF1 resulted in a similar representation of mean standardized CSF protein level as using 2944 proteins in BF2."

I have two questions on these statements. First, the CSF levels in BF1 and CSF levels in BF2 are of different length. How is the correlation coefficient calculated? Second, the latter statement is not very clear. Even with a high correlation coefficient, stating about similar representation may not be accurate or requires further evidence.

- Three general reference proteins (CBLN4, PTPRN2 and PTPRS) were found but are unavailable in BF1. I appreciate the data-driven approach taken by the authors on utilizing t-sne for identifying a cluster to find out the reference proteins. If I replace BF2 with another dataset to repeat this step, I may end up with new set of proteins. Thus, future works on this should not use these reference proteins readily as the clustering via t-sne can vary a lot if we change the data. I would suggest authors to state this limitation in the manuscript.

- Supplementary Figure 8 demonstrates correlation among NTRK3, NTRK2, BLMH and also among CBLN4, PTPRN2 and PTPRS included. Did the authors perform diagnostic such variance inflation factor? Inclusion of these in the regression model may lead to high multi-collinearity which can influence the result.
- The authors used β with different terminology (absolute β coefficients, β -coef, etc.) before properly introducing the notation. Although n is widely used for sample size, however, it is always recommended to introduce the notation first and then use. In the supplementary file, linear and logistic regression models can be introduced using simple notation which would be easier to understand what are β s.
- While reporting p -values, it is suggested to report the values in a standard manner. For example, very small values can be replaced by “<0.001”, “<0.05”, “<0.01”, etc.

We thank the reviewers for the time and effort that they have dedicated to this work, and for the many insightful comments on how to improve this manuscript further. We have considered all the suggestions provided and have incorporated what we think are suitable changes into our new version of the manuscript. Please find the changes in the manuscript/supplementary files and our detailed point-to-point response below (all changes tracked, and responses highlighted in orange font).

Reviewer #2 (Remarks to the Author):

For point 1 and related responses: Thank you for calculating the VIF factors. It remains unclear from these analyses how the VIF values actually relate to these change in beta's displayed in supplementary figure 13: Here some beta's decrease, whereas other beta's increase or remain stable. From the histogram it is not possible to determine whether this may or may not have had an effect. Thus, although it is reassuring that the authors show that that VIF effects at not excessively high for most variables tested, it remains unclear if there is an association between particular beta's and higher VIF.

Authors' response: To address the reviewer's concerns, we created a scatter plot of change in beta-coefficient versus VIF value for all 368 proteins in BF1 for the corresponding analysis, see below. Results showed that for the 4% of proteins with VIF value > 5 , adjusting for the reference protein NTRK3 always resulted in a decreased beta-coefficient. Furthermore, 10 of the 15 proteins with VIF values > 5 were part of cluster 11, with the highest VIF value for the other reference protein candidate NTRK2 (VIF=13.0). As the proteins of cluster 11 were selected to represent the same signal (inter-individual average CSF protein variability) in the BioFINDER-2 training set, it is positive that these proteins show high correlation with NTRK3 also in BioFINDER-1, further confirming the generalizability of the reference proteins.

The alternative approach that first regresses out the reference protein from the outcome, is indeed not identical to directly including both reference and SNP in the same model, but the idea remains the same, i.e., to test what part of the variance in the outcome is explained by the SNP after correcting within the model (assuming that type III variance testing was performed here), or before. Yes, it is clear that there is a specific group of proteins that seems to correlate with each other, with disease processes and with the genetic variant. The present analyses indicate that the correlations do not lead to excessive VIF values, but to state that thus the relationship between a genetic variant and protein levels as outcome are false cannot be concluded from the analyses (see next point).

For point 2 and related responses: From a conceptional point of view the interpretation of these analyses remains flawed, and is not resolved with the new analyses that the authors present: the reference proteins were selected to be correlated with p-tau levels but their mechanistic relationship with the GMNC-OST SNP remains unclear and cannot be determined based on the present data. A confounding variable should influence both the outcome as well as the other predictor. However, in this case the other predictor is a genetic variant, it cannot be assumed that the reference protein levels would influence the genetic variant, as a true confounding variable should do. Other alternative explanations may be conceptually more likely: either this genetic variant influences the levels of both the outcome as well as the references proteins directly or indirectly via another mechanism that is related to all proteins tested here, i.e., the relationship between reference and outcome protein are both downstream effects of the genetic variant, or alternatively, that the reference protein levels themselves are influenced by the mechanism related to tau tangling (and thus p-tau levels) as well as by the genetic variant, which would indicate collider bias.

Authors' response: There seems to be some miscommunication. The “alternative explanation” the reviewer brings up (“the relationship between reference and outcome proteins are both downstream effects of the genetic variant”) is consistent with what we believe is happening. We understand the reviewer’s semantic argument about the statistical definition of a confound, since this associational analysis has no causal directionality. As mentioned many times before, this semantic argument does not change any of our results – we still must modify our interpretations of previous work (including our own) in light of these new findings. However, to ensure the text is as accurate as possible, we have made the following changes based on feedback from the Reviewer:

Lines 120-122: “Consequently, our overarching aim was to establish the concept of non-disease related inter-individual differences in CSF protein levels, and to search for optimal reference protein candidates that could be used to account for this ~~confounding~~ variability in a robust way.”

Lines 356-358: “We conclude that many proteins are affected by individual CSF levels, including proteins relevant in AD, and may benefit from adjustment of this ~~confounding~~ factor when used as biomarkers.”

Lines 504-505: “We show that non-disease related inter-individual differences in CSF protein levels ~~confounds~~ affects diagnostic and prognostic performance for several CSF biomarkers.”

Lines 928-930: “The further right the protein is located, the more associated with the mean standardized CSF level and therefore more strongly ~~confounded by~~ co-varying with the mean standardized CSF protein level ~~when used as a biomarker.~~”

We would also like to highlight that our results clearly speak against the “collider bias” hypothesis, as we show in Supplementary_Reference_Candidates that the reference proteins were not associated with tau tangles in the brain. The reference protein candidates showed AUCs of only 0.62-0.64 when predicting tau PET positivity.

Although, there is speculation on potential mechanisms, such as the role of ventricular volume, those mechanisms are not tested in the present manuscript.

Authors’ response: The association between ventricular volume and pQTLs in the GMNC-OSTN region has been published in a previous work from our group using the same BioFINDER-1 data (see Hansson et al., 2023). There, it was seen that when adjusting for ventricular volume, β -coefficients were changed by up to 14% (generally decreased). The rationale behind the pQTL analysis was therefore that since the mean standardized CSF protein level was associated with ventricular volume ($\beta = -0.321$, $P < 0.001$), we suspected (and confirmed) that the tested genetic variant was more related to the signal of mean standardized CSF protein levels rather than single individual proteins. However, as we also showed that ventricular volume only explained part of the variability in the mean standardized CSF protein level, other CSF properties likely explain why the changes in β -coefficients when adjusting for a reference protein often were a lot larger than 14%. We encourage future works to further explore these mechanistic CSF properties.

Given that the VIF were indeed not excessively high for most proteins in supplementary figure 13, another possibility could also be that these proteins adjusted for the disease effect, since these genetic correlations were tested across the group with both controls and AD individuals, ‘correcting’ for a variable related to tau (and possibly amyloid pathology) would correct for the disease effect on correlated proteins.

We want to highlight that in the first revision, we performed the same analysis as in Supplementary Table 8 for cognitively unimpaired A β -negative participants (i.e. in healthy people) with congruent results. In addition, we have now performed a similar analysis as in Supplementary Fig 13, but with only cognitively unimpaired A β -negative participants, also with congruent results:

Effects of the most common CSF *trans*-pQTL variant rs71635338 in associations with CSF proteins without and with reference protein NTRK3. The SNP rs71635338 is evaluated by association to each CSF OLINK protein in BF1 (n=368) in a linear regression model, without and with adjustment for the reference protein NTRK3. The difference is shown by β -coefficient (left) and P-value (right) for every protein. In general, the associations were severely weakened when adjusting for a reference protein. All models included only cognitively unimpaired A β -negative BF1 participants (n=1445) and were adjusted for age, sex, dementia diagnosis and ten genetic principal components.

The authors present in their rebuttal an ANOVA analyses, comparing the mean Z-scores across proteins in individuals between diagnostic groups, and the MCI, AD and OD groups show on average much larger levels than controls (there were no p values to be found in my version of the rebuttal; but given that these are Z scores there is almost a standard deviation difference between controls and MCI/dementia), which is quite a lot. It is unclear why this comparison is made based on clinical status, while table 1 in the main manuscript indicate that roughly half the SCD and MCI groups are amyloid positive and the other amyloid negative, similarly for tau status. The reference proteins and average Z scores were related to those pathologies so why lump negative and positive together?

Authors' response: We are happy to clarify. This analysis presented P-values and not mean Z-score differences, highlighting how there was no significant difference between diagnostic groups. Therefore, large values close to 1 did not imply almost a standard deviation difference; on the contrary, a large P-value indicated a very small difference between the two groups. We have now added the mean Z-score difference between the groups in this analysis (both before and after regressing out age and sex), see below. The Z-score differences are close to 0 for all cases, with the maximal absolute difference being 0.20 before regressing out age and sex, and 0.12 after.

See also the addition of cognitively unimpaired (CU, including NC and SCD) amyloid-negative individuals vs cognitively impaired (CI, including MCI and AD) amyloid-positive individuals.

No significant difference was seen between any groups.

The mean standardized CSF level did not significantly differ between diagnostic groups.

Mean Z-score difference and *P*-values (adjusted for age and sex and FDR corrected) assessed by a one-way ANCOVA comparing differences in mean standardized CSF protein levels between diagnostic groups. This analysis included BF2 training set participants, split by diagnosis: normal cognition (NC), subjective cognitive decline (SCD), mild cognitive impairment (MCI), Alzheimer dementia (AD) or another neurodegenerative disease (OD). Cognitively unimpaired (CU, including NC and SCD) amyloid-negative individuals were also compared against cognitively impaired (CI, including MCI and AD) amyloid-positive individuals.

Mean standardized CSF protein level	NC vs SCD	NC vs MCI	NC vs AD	NC vs OD	SCD vs MCI	SCD vs AD	SCD vs OD	MCI vs AD	MCI vs OD	AD vs OD	CU ^{AB-} vs CI ^{AB+}
Z-score difference	-0.20	-0.13	-0.18	-0.10	0.07	0.016	0.10	-0.06	0.03	0.09	0.09
Z-score difference after regressing out age and sex	-0.12	-0.02	-0.03	-0.02	0.10	0.09	0.10	-0.01	0.00	0.01	0.01
P-value	0.12	0.90	0.72	0.93	0.20	0.12	0.12	0.73	0.93	0.66	0.60

Thus, from those additional analyses it cannot be concluded that there is no disease effect on the average protein levels (which happen to correlate with tau and/or amyloid?). The models presented based this data cannot distinguish whether the reference proteins are indeed true confounders, colliders or are all downstream effects of the genetic variant. Which brings me back to the issue previously raised with these analyses, which are indeed not the main analyses in the present manuscript, but still these results are used by the authors to make generalising implications that per default all CSF proteomic analyses require correction with reference proteins. This suggestion remains unsupported by the present data. Whether or not to adjust analyses for a particular confounder should be determined on the hypothesis tested and not based on these analyses only. Therefore, the conclusion provided by the authors remains unsupported by the data and analyses performed, in particular the genetic part: “We show that non-disease related inter-individual differences in CSF protein levels confounds diagnostic and prognostic performance for several CSF biomarkers. These differences can also result in false conclusions regarding associations between different CSF proteins or their relations to genetic variations. The issue can be addressed by using certain CSF reference proteins to represent the non-disease related concentration of the protein studied.” What can be concluded based on the present analyses is that certain genetic variants may influence the levels of specific groups of proteins. Whether such associations are ‘false’ cannot be based on these analyses. Indeed, if these analyses were not part of the main objectives, and causality cannot still not be inferred with the present data, then why present these results in such a way?

Authors’ response: We agree with the reviewer that the term confounder should be removed (see response above). Still, we are confident that our analyses show that mean

differences in CSF protein levels exist and if it not adjusted for, it is possible that this signal explains a considerable part of variation in CSF biomarkers and are driving relationships in statistical models. This is an important phenomenon to be aware of in the field of CSF biomarkers and we consider our conclusions supported by the analyses in this work.

For point 3 and related responses: The authors argue that reference proteins are representative of inter-individual variation in CSF protein concentrations, seems circular: that results remain comparable when just taking the Z score average of all protein levels instead of a single protein in the present data set, but the reference protein with about a third of proteins had a similar correlation pattern across individuals and thus influence the average Z-score (they are not independent measurements). Most relationships are attenuated with average Z-score, which may reflect that this indeed also includes proteins that do not correlate with e.g., tau, unlike the selected reference proteins. It remains unresolved what the underlying mechanism may be other than that a specific group of proteins changes similarly across the disease.

Authors' response: The purpose of a reference protein is indeed to represent the inter-individual variation in CSF protein concentrations (which we did not see “changes similarly across the disease” as the reviewer mentions). Additionally, this signal is present in almost all highly expressed proteins in the CSF (not only one third), which further emphasizes its dominance (see Supplementary Fig. 4 below, where all proteins with low detectability have been filtered out).

From a practical perspective, it is of course easier to measure the concentration of a single protein instead of almost 3000 to create an individual mean CSF protein level, which is why we searched for a single candidate (reference protein) to represent this signal. This rationale has been extensively discussed in the manuscript as well as in previous revisions.

Supplementary Figure 4: Visualization of individual CSF levels only including proteins with missing frequency < 75%. For each participant (row), the z-score of 1730 highly detected CSF proteins, sorted by absolute association with mean CSF level, is displayed. As expected, the subject-specific CSF level is prominently shown in highly expressed proteins.

To further clarify the point on APOE, which was only a detail: APOE was taken as an example to point out that it is not a trivial task to determine whether protein levels may or may not be disease related, which in this manuscript was based on having normal cognition and no AD pathology. APOE genotype has effects on CSF protein levels even before people have dementia, and indeed even before people have abnormal amyloid. According to the rationale presented in this manuscript, where non-disease related is based on finding altered protein levels in controls as well, this would disqualify APOE as being related to the disease (which clearly is related to the disease, which the authors also acknowledge).

Authors' response: In this work, we do not make new definitions for specific proteins as disease or non-disease related. The term non-disease related is used in the context of the signal of average CSF protein levels, which we show exists independent of diagnostic group and the neurodegenerative diseases that are present in our cohort.

Hansson, O., Kumar, A., Janelidze, S., Stomrud, E., Insel, P. S., Blennow, K., Zetterberg, H., Fauman, E., Hedman, Å. K., Nagle, M. W., Whelan, C. D., Baird, D., Mälarstig, A., & Mattsson-Carlgen, N. (2023). The genetic regulation of protein expression in cerebrospinal fluid. *EMBO Molecular Medicine*, 15(1).
<https://doi.org/10.15252/emmm.202216359>

Reviewer #4 (Remarks to the Author):

The authors address an important issue of non-disease related inter-individual variability in cerebrospinal fluid (CSF) protein levels and its potential impact on the diagnostic accuracy of Alzheimer's disease (AD) CSF biomarkers. The authors provide a data driven approach to identify potential reference proteins that can significantly improve the accuracy of AD biomarkers. I have a few questions or comments for further clarification on statistical application.

Comments:

If I understood properly, mean standardized CSF level was computed as the average z-score over all 2,943 + A β 40 for BF2 and all 368 OLINK proteins + A β 40 for BF1. For researchers who are going to work on data like this, having a clear picture would be helpful. For instance, letting y_i as the dichotomous response and denoting $z_1, z_2, \dots, z_{np_{BF2}}$ as the z-scores corresponding to 2943 proteins + A β 40 for BF2, where number of proteins, $c = 2944$. Then, mean standardized CSF for the i th subject is calculated as $\frac{1}{np_{BF2}} \sum_{j=1}^{np_{BF2}} z_j$. Similar or convenient notation can be used for BF1 (e.g., $np_{BF2} \rightarrow np_{BF1} = 369$).

Authors' response: We thank the reviewer for recommending this way of denoting the mean standardized CSF protein level. We agree that this way makes the manuscript clearer and will enable other researchers to reproduce the mean standardized CSF level more easily in their own data. In the manuscript, we have therefore now updated lines 156-158 in the results section:

“A mean standardized CSF protein level was calculated for each participant, representing how many standard deviations the concentration of CSF proteins in that participant deviated from the population mean (details in Methods: Statistical analyses, Eq. 1).”

and lines 593-597 in the methods section:

“For every participant i , a mean standardized CSF level y_i was computed as the average z-score over all 2,943 OLINK proteins + A β 40 for BF2, and all 368 OLINK proteins + A β 40 for BF1, Eq. 1:

$$y_i = \frac{1}{n_{proteins}} \sum_{j=1}^{n_{proteins}} z_j. \quad (1)$$

where z_j is the z-score of protein j . Consequently, $n_{proteins}=2944$ in BF2 and $n_{proteins}=369$ in BF1.”

Here $np_{BF2} \ll np_{BF1}$, in this case, could the author justify BF1 data represent similar structure as for BF2?

“The correlation coefficient between a mean standardized CSF level created from 1) all 2944 proteins and 2) a subset of the 369 proteins that were present in BF1, was 0.964 in BF2. This suggests that the subset of 369 proteins measured in BF1 resulted in a similar representation of mean standardized CSF protein level as using 2944 proteins in BF2.”

I have two questions on these statements. First, the CSF levels in BF1 and CSF levels in BF2 are of different length. How is the correlation coefficient calculated? Second, the latter statement is not very clear. Even with a high correlation coefficient, stating about similar representation may not be accurate or requires further evidence.

Authors’ response: Thanks for these important comments. For this analysis, we only used the BF2 cohort (not BF1). As all the 369 proteins that existed in BF1 also existed in BF2, we created a similar subset of 369 proteins in BF2 and used it to simulate a mean standardized CSF protein level with fewer proteins (corresponding to how was done in BF1). In BF2, we then had two ways of calculating a mean standardized CSF protein level for each participant: using all 2944 proteins or using only the overlapping 369 proteins. These two vectors were of same length as they both only included BF2 participants, from which we could calculate a correlation coefficient and saw that the representations were very similar (correlation coefficient of 0.964).

It is indeed the highlighted section by the reviewer that was aimed to explain this analysis and justify that the different number of proteins in BF1 and BF2 did not have a large effect on the interpretation of the mean standardized CSF protein level. However, we realize that this could probably have been written in a clearer way and have therefore adjusted the wording in the manuscript, lines 598-604:

“We investigated if a mean standardized CSF protein level calculated from the subset of 369 proteins (as was done in BF1) represented a similar structure as one that was calculated from 2944 proteins (as was done in BF2). In BF2, we computed a mean standardized CSF protein level for each participant using both the full sample (2944) and subsample (369) of proteins. The correlation coefficient between these two mean standardized CSF protein levels was 0.964, suggesting that the subset resulted in a similar representation of mean standardized CSF protein level as using all 2944 proteins.”

Three general reference proteins (CBLN4, PTPRN2 and PTPRS) were found but are unavailable in BF1. I appreciate the data-driven approach taken by the authors on utilizing t-sne for identifying a cluster to find out the reference proteins. If I replace BF2 with another dataset to repeat this step, I may end up with new set of proteins. Thus, future works on this should not use these reference proteins readily as the clustering via t-sne can vary a lot if we change the data. I would suggest authors to state this limitation in the manuscript.

Authors’ response: We agree with the reviewer that the data-driven method used in this manuscript makes it highly probable that for a new dataset, a corresponding t-SNE plot would visually look very different. Cohort properties could potentially also alter the distance between certain proteins (and subsequently clustering) if their pairwise relationships are not preserved between cohorts.

However, we want to highlight that the actual clustering of proteins, and thereby search for reference proteins, was not completely data-driven. Instead, this was a semi-supervised approach in the sense that we aimed to “capture” an area in the t-SNE plot with certain reference protein characteristics, which were highlighted with yellow colors in Fig. 3e-3h (see below). These reference protein characteristics included associations with the mean standardized CSF protein level, ventricular volume, and main predictors CSF P-tau181/A β 42 in A β -negative individuals, as well as improvement of prediction models P-tau181 \rightarrow TauPET and A β 42 \rightarrow A β PET. Therefore, even if the t-SNE plot may appear visually different when using other datasets, the set of proteins that are highlighted and clustered using this approach will remain similar as long as these proteins also express the reference protein characteristics in the other datasets. All reference proteins performed well in the left-out BF2 test set. Additionally, from the external validation in BF1, we saw that three of the four available reference proteins (A β 40, NTKR3 and NTKR2) improved the models. This confirms that the semi-supervised approach led to generalizable reference proteins.

We have followed the reviewer’s advice and commented on this also in the manuscript, lines 490-494:

“Furthermore, t-SNE is a data-driven method that with high probability would generate a very different visual result for a new dataset, limiting reproducibility. However, the actual clustering of proteins was semi-supervised (based on reference protein characteristics) and generated generalizable reference proteins, which we were able to confirm using a left-out BF2 test set and the external, independent cohort BF1.”

FIGURE 3: Dimensionality reduction reveals a cluster of CSF proteins with desired reference protein characteristics. T-distributed stochastic neighbor embedding (t-SNE), reducing the high dimensional space of 658 participants to a two-dimensional one. Each scatter point illustrates one of the 2,944 CSF proteins, with relative similarity of pairwise proteins aimed to be preserved. In c)-h), min-max scaling for six different criteria has been performed to visualize relative differences within the space, all plots ranging between 0-1 (dark blue to yellow). In c)-f) associations are analyzed as absolute β -coefficients, all adjusted for age and sex. **a)** The raw t-SNE map. **b)** Semi-supervised K-means (K=20) clustering of t-SNE map, aiming to separate the evident clusters from t-SNE dimensionality reduction and areas of overlap in c)-h). t-SNE map colored by **c)** absolute association with mean standardized CSF level; **d)** absolute association with ventricle volume; **e)** absolute association with biomarker CSF P-tau181 (cognitively unimpaired A β -negative participants only); **f)** absolute association with biomarker CSF A β 42 (cognitively unimpaired A β -negative participants only); **g)** model performance when used as reference protein in **P-tau181**→**TauPET**; **h)** model performance when used as reference protein in **A β 42**→**A β PET**.

Supplementary Figure 8 demonstrates correlation among NTRK3, NTRK2, BLMH and also among CBLN4, PTPRN2 and PTPRS included. Did the authors perform diagnostic such variance inflation factor? Inclusion of these in the regression model may lead to high multi-collinearity which can influence the result.

Authors' response: We would like to point out that the proteins in Supplementary Fig. 8 are the selected reference protein candidates, all of which we selected to represent the same confounding inter-individual CSF variability signal. Therefore, as expected, these proteins are highly correlated. We only used one of them at a time in each statistical model (avoiding potential collinearity). For a more robust measure of this signal, created from several reference proteins, we used the first component of a singular value decomposition, again represented in a single independent variable (avoiding potential collinearity).

The authors used β with different terminology (absolute β coefficients, β -coef, etc.) before properly introducing the notation. Although n is widely used for sample size, however, it is always recommended to introduce the notation first and then use. In the supplementary file, linear and logistic regression models can be introduced using simple notation which would be easier to understand what are β s.

Authors' response: We have followed the reviewer's advice and added simple descriptions and equations of the linear and logistic regression models in the Supplementary Methods file:

Statistical models

Linear regression

A linear regression model assumes a linear relationship between a dependent variable y and independent variables x_n :

$$y_i = \beta_0 + \sum_{n=1}^N \beta_n x_{in} + \varepsilon_i. \quad (1)$$

where i is one observation, N is the number of independent variables, β_0 is the intercept and ε is the error term. In this work, β -coefficients were used as a metric for strength (absolute value) and direction (sign) of an association between standardized variables.

Logistic regression

A logistic regression model assumes a linear relationship between the log-odds of an event and independent variables x_n :

$$p(x_i) = \frac{1}{1 + \exp(-(\beta_0 + \sum_{n=1}^N \beta_n x_{in}))}. \quad (2)$$

where p is the probability, i is one observation, N is the number of independent variables, β_0 is the log odds when all independent variables are 0. β -coefficients explain change in log-odds for a 1-unit difference in corresponding x_n .

We refer to this addition in the methods section, lines 628-632:

*“To analyze associations with a continuous dependent variable, linear regression models (**Supplementary Methods: Statistical models**) were applied. In addition, partial correlation coefficients (Pearson) were computed to study correlation matrices between continuous variables. To predict a dichotomous variable, logistic regression models were applied (**Supplementary Methods: Statistical models**).”*

While reporting p -values, it is suggested to report the values in a standard manner. For example, very small values can be replaced by “<0.001”, “<0.05”, “<0.01”, etc.

Authors’ response: We appreciate the Reviewer’s perspective, but we believe reporting the actual p -values is more transparent. Additionally, some analyses include relationships where p -values are expected to be a lot smaller than 0.001 (for example in pQTL analyses) and it therefore would not be sufficient to report <0.001. Therefore, we have opted not to make this change.

REVIEWERS' COMMENTS

Reviewer #4 (Remarks to the Author):

Reviewer's report for "Cerebrospinal fluid reference proteins increase accuracy and interpretability of biomarkers for brain diseases"

I would like to thank the authors for responding my comments or concerns. I provide few additional comments. Reading the response letter, I would like to share some comments on multicollinearity and causality to help readers understand the work.

Comments related to multicollinearity and causality:

- The variance inflation factor (VIF) is a measure to investigate multicollinearity in the regression analysis. I want to point out that the change in betas (some decreased while others increased or remained stable) is not directly related to VIF. When we include additional covariate (e.g., reference protein), the relationship between outcome and the existing covariates may change. Even if the beta coefficients become smaller, VIF may increase or vice versa. It is because the VIF values do not directly correspond to the magnitude of beta coefficients in the regression model (where main outcome is regressed on the covariates).

The absence of excessively high VIF values suggests that multicollinearity is not a major concern in the model as the authors demonstrated in this manuscript. Even with low VIF values, correlations between predictors may still exist without violating the assumption of collinearity. Nonetheless, the use of VIF remains valuable and widely used for detecting and addressing multicollinearity, enhancing the reliability of regression analyses.

- I found the authors' goal is not to explore the causal relationship. Thus, the wording of confounding may confuse readers unless this research show evidence of causality. Reporting and interpreting the results of linear regression assume association not causation unless otherwise the goal and relevant assumptions are taken care of for causal inference. I suggest authors to mention in the manuscript that causation is not the focus which may require further investigation and please be cautious while using the terms such as confounders.

In Supplementary Figure 13, the title can be slightly misleading (such as .. "In general, the associations were severely weakened when adjusting for a reference protein" ..). Instead it is always better to state using summary statistics (average or median, etc.).

Minor comment:

- In both linear and logistic regression, the subscript i represent i th subject which can vary from 1 to the number of samples available.

Reviewer's report for "Cerebrospinal fluid reference proteins increase accuracy and interpretability of biomarkers for brain diseases"

I would like to thank the authors for responding my comments or concerns. I provide few additional comments. Reading the response letter, I would like to share some comments on multicollinearity and causality to help readers understand the work.

Comments related to multicollinearity and causality:

- The variance inflation factor (VIF) is a measure to investigate multicollinearity in the regression analysis. I want to point out that the change in betas (some decreased while others increased or remained stable) is not directly related to VIF. When we include additional covariate (e.g., reference protein), the relationship between outcome and the existing covariates may change. Even if the beta coefficients become smaller, VIF may increase or vice versa. It is because the VIF values do not directly correspond to the magnitude of beta coefficients in the regression model (where main outcome is regressed on the covariates).

The absence of excessively high VIF values suggests that multicollinearity is not a major concern in the model as the authors demonstrated in this manuscript. Even with low VIF values, correlations between predictors may still exist without violating the assumption of collinearity. Nonetheless, the use of VIF remains valuable and widely used for detecting and addressing multicollinearity, enhancing the reliability of regression analyses.

Author's response: We agree with reviewer on their description of VIF and that the absence of excessively high VIF values importantly suggests that multicollinearity is not a major concern in our models.

- I found the authors' goal is not to explore the causal relationship. Thus, the wording of confounding may confuse readers unless this research show evidence of causality. Reporting and interpreting the results of linear regression assume association not causation unless otherwise the goal and relevant assumptions are taken care of for causal inference. I suggest authors to mention in the manuscript that causation is not the focus which may require further investigation and please be cautious while using the terms such as confounders.

Author's response: We thank the reviewer for highlighting this and agree that this is an important point. We have removed the term "confounding" throughout the manuscript and, as suggested by the reviewer, have added a sentence regarding causation in the manuscript discussion (lines 496-498):

"We also want to highlight that statistical associations in cross-sectional data were the focus in this work. Conclusions about causal relationships would require further investigation."

- In Supplementary Figure 13, the title can be slightly misleading (such as .. "In general, the associations were severely weakened when adjusting for a reference protein"..). Instead it is always better to state using summary statistics (average or median, etc.).

Authors' response: We agree with the reviewer that summary statistics is more appropriate in this context. In responding we found a small error in the plot of this figure. We have fixed the error and re-plotted it here. This does not change interpretation in any way. We replaced the line “In general, the associations were severely weakened when adjusting for a reference protein” in the figure legend, see below:

Supplementary Figure 13: Effects of the most common CSF trans-pQTL variant rs71635338 in associations with CSF proteins without and with reference protein NTRK3. The SNP rs71635338 is evaluated by association to each CSF OLINK protein in BF1 (n=367, all except NTRK3) in a linear regression model, without and with adjustment for the reference protein NTRK3. The difference is shown by β -coefficient (left) and two-sided P-value (right) for every protein. **On average, β -coefficients decreased from 0.088 to 0.012, and P-values increased from 0.12 to 0.41 when adjusting for a reference protein.** All models included BF1 participants (n=1445) and were adjusted for age, sex, dementia diagnosis and ten genetic principal components. A variance inflation factors (VIF) analysis revealed that the models did not violate collinearity assumptions, as 96% of the proteins had VIF <5.

Minor comment:

- In both linear and logistic regression, the subscript i represent i th subject which can vary from 1 to the number of samples available.

Authors' response: In both the linear and logistic regression model description (Supplementary methods) the description of subscript i has now been adjusted accordingly:

“for the i th observation where i varies from 1 to the number of samples available (in our case usually number of participants)”